# Active Power Loss Reduction for Radial Distribution Systems by Placing Capacitors and PV Systems with Geography Location Constraints

**Thuan Thanh Nguyen** [1] , **Bach Hoang Dinh** [2,*] , **Thai Dinh Pham** [3] **and Thang Trung Nguyen** [2]

1   Faculty of Electrical Engineering Technology, Industrial University of Ho Chi Minh City, Ho Chi Minh City 700000, Vietnam; nguyenthanhthuan@iuh.edu.vn
2   Power System Optimization Research Group, Faculty of Electrical and Electronics Engineering, Ton Duc Thang University, Ho Chi Minh City 700000, Vietnam; nguyentrungthang@tdtu.edu.vn
3   Institute of Research and Development, Duy Tan University, Danang 550000, Vietnam; phamdinhthai@duytan.edu.vn
*   Correspondence: dinhhoangbach@tdtu.edu.vn

**Abstract:** This paper presents a highly effective method of installing both capacitors and PV systems in distribution systems for the purpose of reducing total power loss in branches. Three study cases with the installation of one capacitor, two capacitors and three capacitors were implemented and then the optimal solutions were used to install one more photovoltaic (PV) system. One PV system with 20% active power of all loads and less than active power of all loads was tested for two different conditions: (1) with geography location constraint and (2) without geography location constraint for PV system placement. The results from two systems consisting of 33 and 69 nodes were obtained by using the Stochastic Fractal Search Optimization Algorithm (SFSOA). Simulation results show that this method can determine the appropriate location and size of capacitors to reduce the total power losses more effectively than other existing methods. Furthermore, the paper also demonstrates the real impact of using both capacitors and PV systems to reduce active power loss as well as improve the voltage profile of distribution systems. This paper also finds that if it is possible to place PV systems in all nodes in distribution systems, the benefit from reducing total loss is highly significant and the investment of PV system placement is highly encouraged. As a result, it is recommended that capacitors and PV systems be used in distribution networks, and we claim that two important factors of the installed components consisting of location and size can be determined effectively by using SFSOA.

**Keywords:** photovoltaic systems; capacitor placement; PV system placement; distribution network; active power loss; voltage profile

## 1. Introduction

In radial distribution networks, shunt capacitors should be installed at appropriate places and in appropriate sizes for reducing currents flowing in branches, improving voltage profiles and minimizing total power losses [1–3]. As generally estimated, power loss of distribution networks takes about 13% of total generated power of power systems [4] and it can be reduced by installing shunt capacitors to decrease reactive power flows [5]. In addition, voltage drops can also be improved by reducing the branch currents. Thus, installing capacitors plays a very important role in reducing total power losses and improving the voltage profile of distribution networks. In fact, appropriately selecting the number of capacitors and the location and capacity of individual capacitors are very important issues

in designing and operating distribution networks [6–10] where the higher the number of capacitors is installed, the better of voltage profile is, but also the higher investment and operating costs we have to pay.

In recent decades, many optimization methods, including deterministic approaches, conventional meta-heuristic and modified versions of meta-heuristic approaches as well as hybrid meta-heuristic approaches have been applied for optimally installing shunt capacitors to reduce total power losses. In some of the earliest studies [11–14], an analytical approach was applied for small scale radial distribution networks from 15 nodes to 33 nodes. The methods employed two steps in which the first step was to determine the best suitable nodes to place capacitors and the second step was to calculate the appropriate values of reactive power generation for each placed capacitor. These methods can be classified as deterministic methods because their results seemed not to change through different runs. Even though the deterministic approach could effectively reduce total power losses of the considered networks, the disadvantage is that it is only capable of being applied in small scale networks. To overcome this drawback, an alternative trend of using meta-heuristic methods has been developed for the optimal placement and size of capacitors in distribution networks. Two methods, Particle Swarm Optimization (PSO) and Genetic Algorithm (GA), accompanied by their modified versions [15–20] are most popular approaches for this optimization problem. The PSO based methods, including PSO with constriction and inertia weight factors (CIF-PSO) [15], PSO with Multiple Agents (MAs-PSO) [16] and PSO with different applied distributions (DADs-PSO) [17] have been successfully applied, where the best capacitor locations were found according to a factor of loss sensitivity in CIF-PSO [15] and a fuzzy inference in MAs-PSO [16]. Such methods could be effectively applied for large scale distribution networks up to 85 nodes and could reach a significant reduction in losses. Another study in [17] dismissed the step of determining best capacitor locations by applying DADs-PSO methods to reduce the calculation time. However, its performance was still in question. In fact, only one small-scale system with 10 nodes was employed to test these methods, and only the Tabu search algorithm (TSA) and conventional GA were shown to have comparable performance. Similarly, GA-based methods including conventional GA [18,19] and modified versions have been introduced where the GA was tested in two small scale systems with 23 nodes and 33 nodes and compared with only PSO based methods and bare systems without compensation. As compared to GA, Real Code-Based GA (RCGA) [20] was applied for more complex cases with 15, 34 and 69-node networks, but its performance was only compared to bare networks without capacitors. Generally, GA and PSO did not show good results for the radial distribution networks with the presence of shunt capacitors.

In recent years, many different algorithms have been applied for this consideration problem, including Mixed Integer Nonlinear Programming Algorithm (MINPA)[21], Combined Practical Algorithm (CPA) [22], Teaching-Learning Algorithm (TLA) [23], Bacterial Foraging Optimization Algorithm (BFOA) [24], Gravitational Search Algorithm (GSA) [25], Flower Pollination Algorithm (FPA) [26,27], Cuckoo Search Algorithm (CSA) [28], Intersect Mutation Differential Evolution (IMDE) [29], Moth Swarm Algorithm (MSA) [30], Power Loss Sensitivity Factor Based Analytical Approach (PLSF-AA) [31], Network Feature Based Heuristic Algorithm (NFBHA) [32] and Interchange Improved Algorithm (IIA) [33]. Among these methods, those that are not based-metaheuristic algorithms need the configuration features to select capacitor locations and compute the reactive power capacities. Such methods include MINPA [21], CPA [22], PLSF-AA [31], NFBHA) [32] and IIA [33], but they are not capable of applying to all power systems, especially for complex systems with a high number of nodes and radial branches. In [21], the problem of capacitor placement in radial distribution networks was formulated as a nonlinear program and solved by a General Algebraic Modeling System (GAMS) software. It could find better locations and more appropriate values of capacitors than other methods, such as PSO and Fuzzy. However, its effectiveness is not persuasively demonstrated due to the lack of comparisons. Similarly, PLSF-AA [31] has used a power loss sensitivity factor to select the location of capacitors and reactive power flow in each branch to determine the capacity of reactive power sources at compensated nodes. In this group, implementing of methods is closely dependent

on the configuration of consideration systems and becomes ineffective for a power system with more complex configuration. Conversely, the implementation of metaheuristic algorithms is simpler and not dependent on different configurations of power systems. They have the advantage by neglecting the first stage of determining capacitor locations as well as being independent from the network's configuration. However, these metaheuristic algorithms have not been persuasively demonstrated for large scale systems in practice, because their searching time has not been mentioned in the studies. All previous studies about capacitor placement are summarized in Table 1.

**Table 1.** Summary of previous studies about capacitor placement for power loss reduction.

| Family Method | Method, Published Year | Study Cases |
|---|---|---|
| Deterministic methods | Two-step method [11], 1999 | 15 and 33-node systems |
| | Two-step method [12], 2008 | 15 and 33-node systems |
| | Two-step method [13], 2013 | 28 and 85-node systems |
| | Two-step method [14], 2015 | 15 and 33-node systems |
| | MINPA [21], 2014 | 10, 34, and 85-node systems |
| | CPA [22], 2015 | 33 and 687-node systems |
| | PLSF-AA [31], 2019 | 33, 69-node systems |
| | NFBHA [32], 2019 | 33, 69 and 119-node systems |
| PSO methods | CIF-PSO [15], 2007 | 10, 15, 34, 69 and 85 bus |
| | MAs-PSO [16], 2013 | 69-node system |
| | DADs-PSO [17], 2015 | 9-node system |
| GA methods | GA [18], 2010 | 22-node system |
| | GA [19], 2016 | 33-node system |
| | RCGA [20], 2008 | 15, 34 and 69-node systems |
| Other metaheuristic algorithms | TLA [23], 2014 | 22, 69, 85 and 141-node systems |
| | BFOA [24], 2014 | 33-node system |
| | GSA [25], 2015 | 33, 69, 85-node systems |
| | FPA [26], 2016 | 10, 33 and 69-node systems |
| | FPA [27], 2018 | 33, 34, 69 and 85-node systems |
| | CSA [28], 2018 | 34 and 69-node systems |
| | IMDE [29], 2016 | 33 and 69-node systems |
| | MSA [30], 2018 | 33 and 69 and 85-nodes systems |
| | IIA [33], 2020 | 33, 34, 69 and 85-nodes systems |

In this paper, Stochastic Fractal Search Optimization Algorithm (SFSOA) [34] is proposed to solve the optimal location and size of capacitors in radial distribution networks. This method is a powerful meta-heuristic algorithm in terms of the accuracy of optimal values because it updates new solutions three times at each search iteration. In fact, the searching strategy of SFSOA performs the diffusion technique to update new locations and new sizes of installation solutions the first time and then continues to extend the search space twice more. The three updating techniques can diversify the searching methods in both local exploitation and global exploration. To investigate the performance of the proposed method, we have applied for two test networks, 33 and 69 nodes, in several installation cases, including one capacitor, two capacitors and three capacitors. After placing capacitors in a distribution network, the searching approach continues to locate one PV system at a node and determine the most appropriate capacity for the highest reduction of total power loss. The simulation results show that the proposed SFSOA method has been successfully applied for optimally determining the location and size of capacitors and PV systems in distribution networks. Thus, in summary, the contributions of the paper are as follows:

(1) Select the appropriate control parameters of SFSOA for finding the best location and the most suitable values for capacitors to reduce total power losses;

(2) Find the best location and capacity of PV systems for reducing total power loss

(3) Demonstrate the fast search time of SFSOA for the considered problem;

(4)   Demonstrate the effectiveness of the placement solutions, only capacitors as well as a combination between capacitors and PV sources in reducing power loss and improving the voltage profile of systems.

In addition to the introduction, the paper has other remaining parts as follows: Section 2 presents the impact of capacitors on power losses, voltage drop and problem formulation. Section 3 presents three main techniques of SFSOA and the impact of factors on the performance of SFSOA. Section 4 shows the implementation of SFSOA for the problem of placing capacitors in radial distribution networks. Section 5 shows obtained results and comparisons. Section 6 summarizes the achievements of the study and includes the conclusion.

## 2. Problem Formulation

### 2.1. The Impact of Capacitors on Power Loss Reduction

For better understanding of the impact of installing capacitors in distribution power networks, we employed a simple example with a two-nodes system shown in Figure 1 where node 0 is a power source and node 1 is an electric load. One distribution line connecting the source and the load has impedance of $Z = R + jX$ ($\Omega$) where $R$ is resistance and $X$ is the reactance. The impedance of the line causes voltage drop and power losses including active power loss ($\Delta P$) and reactive power loss ($\Delta Q$). The level of voltage drop and power losses is mainly influenced by the impedance and power of the load. Basically, the impedance of the line cannot be changed for reducing voltage drop and power losses because the replacement of the line costs a lot of money. Thus, the best way is to vary the transmission power in the line by adjusting the load power, namely the reactive power of the load.

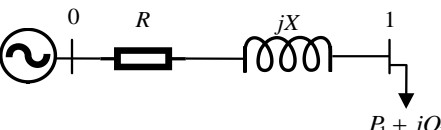

**Figure 1.** A simple distribution system.

Before installing capacitors, the voltage drop, active power loss and reactive power loss, which are represented as $\Delta U^{before}$, $\Delta P^{before}$ and $\Delta Q^{before}$, are determined as

$$\Delta U^{before} = \frac{RP_1 + XQ_1}{U_1} \tag{1}$$

$$\Delta P^{before} = R\left(\frac{P_1^2 + Q_1^2}{U_1^2}\right) \tag{2}$$

$$\Delta Q^{before} = X\left(\frac{P_1^2 + Q_1^2}{U_1^2}\right) \tag{3}$$

where $U_1$ is the voltage of node 1 and Equation (1) is an approximate formula to calculate voltage drop in distribution networks [35].

After installing a capacitor with the capacity $Q_c$, the voltage drop and the power losses, which are represented as $\Delta U^{after}$, $\Delta P^{after}$ and $\Delta Q^{after}$, are calculated by

$$\Delta U^{after} = \frac{RP_1 + X\left(Q_1 - Q_c\right)}{U_1} \tag{4}$$

$$\Delta P^{after} = R\left(\frac{P_1^2 + \left(Q_1 - Q_c\right)^2}{U_1^2}\right) \tag{5}$$

$$\Delta Q^{after} = X\left(\frac{P_1^2 + \left(Q_1 - Q_c\right)^2}{U_1^2}\right) \tag{6}$$

Therefore, the voltage drop and power losses can be reduced by the installed capacitor as shown below:

$$\Delta U^{before} - \Delta U^{after} = \frac{XQ_c}{U_1} \tag{7}$$

$$\Delta P^{before} - \Delta P^{after} = R\left(\frac{2Q_1 Q_c - Q_c^2}{U_1^2}\right) \tag{8}$$

$$\Delta Q^{before} - \Delta Q^{after} = X\left(\frac{2Q_1 Q_c - Q_c^2}{U_1^2}\right) \tag{9}$$

Equations (8) and (9) indicate that the reduction of power losses is dependent on both $Q_1$ and $Q_c$ whereas the reduction of voltage drop is only dependent on $Q_c$. So, if we focus on the reduction of power losses, we should consider the capacitor capacity and the load reactive power. On the contrary, as seen in Equation (7), only the capacitor capacity influences the voltage drop. The higher $Q_c$ is, the higher the voltage drop reduction is. However, if $Q_c$ is higher than the load reactive power, it causes overvoltage and can cause an increase in power losses. In order to clarify the impact of $Q_c$ on the reduction of power losses, we use the result from Equation (8) for further investigation. For the sake of simplicity, the reduction of active power loss is represented as $\Delta P_{reduction}$ and Equation (8) can be rewritten as follows:

$$\Delta P_{reduction} = \frac{R}{U_1^2}\left(2Q_1 Q_c - Q_c^2\right) \tag{10}$$

After taking the derivative of $\Delta P_{reduction}$ with respect to the variable of $Q_c$ and setting obtained results in zero, we have

$$\frac{\partial \Delta P_{reduction}}{\partial Q_c} = \frac{R}{U_1^2}\left(2Q_1 - 2Q_c\right) = 0 \tag{11}$$

The values of $Q_c$ and its impact on the active power loss reduction are shown in Figure 2. This figure shows that $Q_c$ should be set up from zero to $Q_1$ but $Q_c$ is not proportional to the loss reduction. In fact, increasing $Q_c$ leads to reducing the power loss in a range from 0 to $Q_1$ and it approaches the best reduction, i.e., no power loss, at $Q_c = Q_1$. However, such loss reduction is unintentionally increased when $Q_c$ is higher than $Q_1$. So, $Q_c$ should not be higher than $Q_1$.

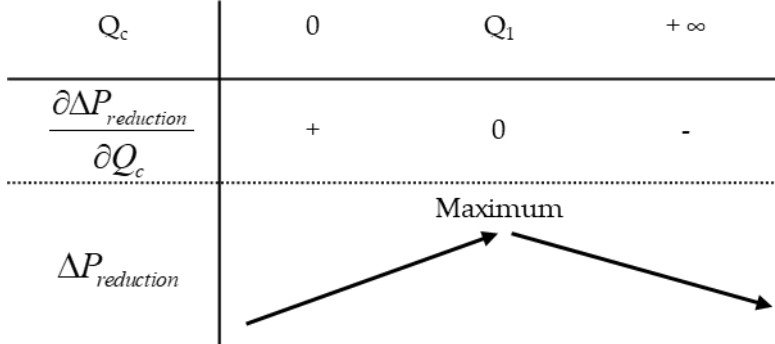

**Figure 2.** The impact of $Q_c$ on active power loss reduction.

Thus, if we install a capacitor with $Q_c = Q_1$, the system can get the best reduction of power loss and the improvement in voltage drop $\Delta U_{reduction}$ can reach the maximum as shown by

$$\Delta U_{reduction} = \frac{XQ_1}{U_1} \tag{12}$$

As analyzed above, the impact of installing capacitors can be summarized as follows:

(1)　Capacitor placement in a distribution power network can reduce power losses and voltage drop;
(2)　The power losses can be minimized if capacitors supply the full reactive power of loads;
(3)　The higher the capacitor's capacity is, the better loss reduction is. However, the reactive power of capacitors should not be higher than the total reactive power of loads;
(4)　The capacitor's capacity is directly proportional to the voltage drop reduction.

### 2.2. Objective Functions

In distribution networks, or the square of branch currents, the distribution power in lines causes the active power loss. The purpose of this study is to determine how to appropriately install capacitors to minimize the active power loss in distribution systems, thus the objective function is the total power losses (TPL) in all branches of the consideration networks. This objective function is mathematically established by

$$\text{Reduce } TPL = \sum_{l=1}^{Nl} \Delta P_l \tag{13}$$

where $\Delta P_l$ is active power loss of the *l*th distribution line and it is calculated as

$$\Delta P_l = 3I_l^2 R_l \tag{14}$$

where $I_l$ is the current flowing in the *l*th distribution line; and $R_l$ is the resistance of the *l*th distribution line.

### 2.3. Constraints

The power quality of distribution networks directly and significantly influences the operation of loads, including industrial and residential customers, especially in industry sectors where the production lines and technology equipment are very sensitive to the variation of power supply. Thus, the working parameters of power supply should be kept in normal ranges called the physical limits and operation constraints of the power system. Those limit conditions of the distribution networks are summarized as follows:

**Current constraint:** Each power line has its own capacity of carrying loads depending on its material type and cross-section area. The overload situation can interrupt power supply by the protective relay and/or damage the line construction, and thus lead to negative economic impact. Consequently, a maximum permissive current of each power line should be set to ensure normal status of lines as shown by the following inequality constraint:

$$I_l^{\max} \geq I_l; \ l = 1, \ \ldots, \ Nl \tag{15}$$

**Voltage Constraint:** Voltage constraint is one of the most important conditions regarding power quality of distribution networks. If supply voltage of nodes has been beyond an acceptable working range, electric devices would fault or operate incorrectly. So, the voltage constraint below should be supervised seriously in the distribution networks:

$$U_m^{\min} \leq U_m \leq U_m^{\max}; \ m = 1, \ldots, Nn \tag{16}$$

where $U_m$ is the working voltage of the *m*th node; $U_m^{min}$ and $U_m^{max}$ are the minimum and maximum working voltage of the *m*th node. Normally, nodes nearby the source node tend to have higher voltage levels than other nodes near the end of the distribution lines.

**Capacitor's capacity:** Reactive power of capacitors injected in the nodes of distribution networks aims to reduce the reactive power provided by the power source at the slack node. Thus, it reduces the currents of distribution lines as well as decreases the power loss of the distribution network. As mentioned previously, the higher the capacitor's capacity is, the better loss reduction is. However, the total reactive power of capacitors must not be too high because this can cause the compensation solution to be ineffective. The total reactive power generation of all capacitors is limited by

$$\sum_{i=1}^{Nc} Q_{ci} \leq Q_c^{\max} \tag{17}$$

where $Q_{ci}$ is the reactive power of the *i*th capacitor; and $Q_c^{max}$ is the maximum capacity of all installed capacitors, which is normally selected as the sum of reactive power of all loads.

## 3. Stochastic Fractal Search Optimization Algorithm (SFSOA)

The search strategy of SFSOA is implemented via three updating phases including the diffusion technique and two other updating mechanisms. In diffusion technique, each old solution is updated *Ndf* times. Thus, in the first phase, there are ($Ndf \times Npo$) newly generated solutions. The two update mechanisms will later produce a maximum of *Npo* new solutions in each phase. As a result, the total number of newly produced solutions is ($Ndf \times Npo + Npo + Npo$) at each iteration. The details of the SFSOA implementation can be described as follows.

### 3.1. Diffusion Technique

This technique was developed from the diffusion phenomenon. It produces Ndf new solutions from each old one by applying the two formulas below:

$$S_x^{new} = \text{normrnd}(S_{best}, std) + \varepsilon \times (S_{best} - S_x) \quad if \quad Rd_x < Walk \tag{18}$$

$$S_x^{new} = \text{normrnd}(S_x, std) \quad if \quad Rd_x \geq Walk \tag{19}$$

where *Walk* is the diffusion factor, which is selected in the range of 0 and 1; $Rd_x$ is a random value within 0 and 1 for the *x*th solution and *std* is the standard deviation between the best solution $S_{best}$ and the *x*th solution, $S_x$. The standard deviation *std* is obtained by

$$std = \left| \frac{log(C_{It})}{H_{It}} \times (S_x - S_{best}) \right| \tag{20}$$

As seen from Equations (19) and (20), *Walk* is a control parameter of the method and its value plays an important role in choosing either Equation (19) or Equation (20) to generate a new solution $S_X^{new}$. If *Walk* is close to 1, there is a very high possibility that the random number $Rd_x$ is less than *Walk* and Equation (19) will be applied. On the other hand, if choosing a very small value of *Walk*, $Rd_x$ is almost never smaller than *Walk* and Equation (20) will be applied. Clearly, if *Walk* is set to 0.5, the possibility of selecting between two equations is the same. Many experiments in three critical values, 0, 0.5 and 1, will be tried for further investigation about the impact of *Walk* on SFSOA performance. The results are shown in Section 5.

### 3.2. The First Update Mechanism

In this update mechanism, not all current solutions can be newly updated as in the diffusion mechanism. At first, all solutions in the population are sorted based on their fitness function values

where the smallest one, i.e., the best quality, is ranked at the top and the highest one, i.e., the worst quality, is ranked at the end. The ranking order of a specific solution $S_x$ is called $P_x$ and $P_x$ is from 1 to $Npo$ where $P_x = 1$ is corresponding to the best solution and $P_x = Npo$ is corresponding to the worst solution. Next, the impact factor of each solution called $IF_x$ is determined by

$$IF_x = \frac{P_x}{Npo} \tag{21}$$

Then, a random number $Rd_x$, between 0 and 1, is produced corresponding to each solution, to compared to $IF_x$. Finally, a newly update condition to decide whether a current solution could get a chance to update its new one or not is described by

$$S_x^{new} = \begin{cases} S_{rd1} - \varepsilon \times (S_{rd2} - S_x) & if\ Rd_x < IF_x \\ S_x & otherwise \end{cases} \tag{22}$$

### 3.3. The Second Update Mechanism

In the third new generating phase using the second update mechanism, all current solutions are updated by using the following formula.

$$S_x^{new} = \begin{cases} S_x - \varepsilon \times (S_{rd1} - S_{best}) & if\ Rd_x > 0.5 \\ S_x + \varepsilon \times (S_{rd1} - S_{rd2}) & otherwise \end{cases} \tag{23}$$

## 4. The Implementation of SFSOA for Placing Capacitors in Radial Distribution Networks

### 4.1. Determination of Control Variables

As analyzed in previous sections, the location and size of capacitors have a significant impact on the considered objective function of total power losses. Thus, in this optimization problem, the location and capacity are defined as the control variables and a solution $S_x$ is a specific set of those control variables as described by

$$S_x = [Po_{i,x}\ Q_{ci,x}];\ i = 1, \ldots, Nc\ \&\ x = 1, \ldots, Npo \tag{24}$$

where $Po_{i,x}$ is the position of the $i$th capacitor corresponding to the $x$th solution and $Q_{ci,x}$ is the reactive power of the $i$th capacitor corresponding to the $x$th solution. The position is varied from the second node to the last node of the distribution network while the capacity of all capacitors must satisfy the constraint of maximum installed capacity as indicated in Equation (17) of Section 2.3.

### 4.2. Determination of the Fitness Function

The fitness function must be defined to reflect the general quality of solutions. In this optimization problem, the fitness function is the sum of total active power losses (i.e., objective function) and penalty terms including the penalties for the violation of the branch currents and the violation of node voltages. Thus, the fitness function of the $x$th solution is as follows:

$$F_x = 3\sum_{l=1}^{Nl} I_{l,x}^2 R_l + \omega_1 \left( \sum_{l=1}^{Nl} (\Delta I_{l,x})^2 \right) + \omega_2 \left( \sum_{m=1}^{Nn} (\Delta U_{m,x})^2 \right);\ x = 1, \ldots, N_{Po} \tag{25}$$

where $\Delta I_{l,x}$ and $\Delta U_{m,x}$ are penalty terms for the violation of the $l$th branch current and the violation of the $m$th node voltage in the $x$th solution [36].

### 4.3. Termination Condition

The search process is implemented until the current iteration $C_{It}$ is equal to the highest number of iterations $H_{It}$.

### 4.4. The Search Process of SFSOA for Optimal Determining the Location and Size of Capacitors in Distribution Networks

The whole search process of the proposed method for determining location and size of capacitors to minimize total active power loss is described in detail as follows:

Step 1: Select values to *Ndf*, *Npo*, $H_{It}$ and *Walk*

Step 2: Randomly produce new solutions

Step 3: Find $I_{l,x}$ and $U_{m,x}$ by running Backward/forward sweep algorithm

Step 4: Calculate $F_x$ using Equation (25)

Step 5: Choose $S_x$ with the smallest value of $F_x$ to be $S_{best}$

-Set $C_{It} = 1$

Step 6: Update population using the diffusion technique presented in Section 3.1.

Step 7: Find $I_{l,x}$ and $U_{m,x}$ by running Backward/forward sweep algorithm

Step 8: Calculate $F_x$ using Equation (25)

Step 9: Compare $S_x$ and $S_x^{new}$ to retain candidate population

Step 10: Find $P_x$ and $IF_x$ using Equation (21)

Step 11: Update population using Equation (22)

Step 12: Find $I_{l,x}$ and $U_{m,x}$ by running Backward/forward sweep algorithm

Step 13: Calculate $F_x$ using Equation (25)

Step 14: Compare $S_x$ and $S_x^{new}$ to retain candidate population

Step 15: Update population using Equation (23)

Step 16: Find $I_{l,x}$ and $U_{m,x}$ by running Backward/forward sweep algorithm

Step 17: Calculate $F_x$ using Equation (25)

Step 18: Compare $S_x$ and $S_x^{new}$ to retain candidate population

Step 19: Choose $S_x$ with the smallest value of $F_x$ to be $S_{best}$

Step 20: If $C_{It} = H_{It}$, stop the process and print results out. Otherwise, increase $C_{It}$ to $(C_{It} + 1)$ and back to step 6.

## 5. Numerical Results

In this section, two test systems including 33 and 69-node radial distribution networks are solved to find the best size of capacitors and the most appropriate nodes for placing the capacitors by applying SFSOA. The configurations of two distribution networks are, respectively, shown in Figures 3 and 4. The total active power and reactive power of all loads are [3715 kW, 2300 kVAr] and [3801 kW, 2695 kVAr] in the first and second distribution networks, respectively. Data of the two test networks are withdrawn from [4,37] and given in detail in Tables A1 and A2 of the Appendix A.

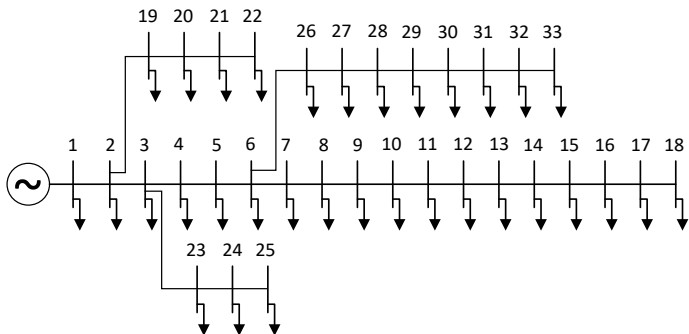

**Figure 3.** The 33-node radial distribution network.

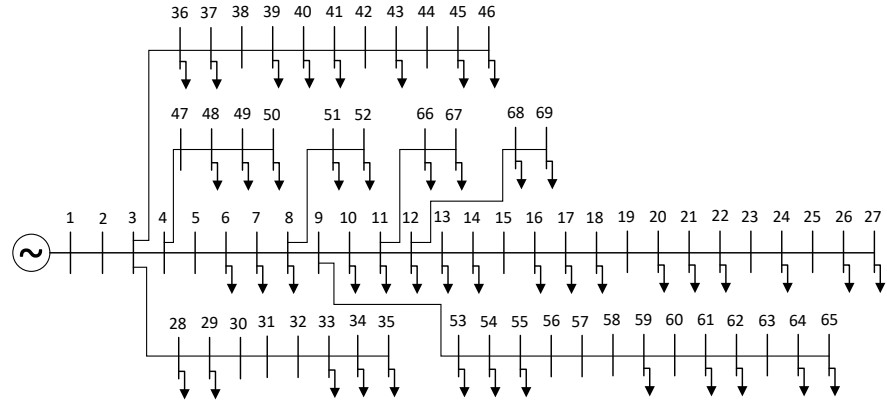

**Figure 4.** The 69-node radial distribution network.

At each network experimented on, three test cases corresponding to three installation options, one capacitor, two capacitors and three capacitors, were sequentially tried and at each test case the investigation covered 50 trial runs to collect the minimum total power losses, the mean total power losses, the maximum total power losses and standard deviation of fifty runs. The simulation was programmed in MATLAB and run on a personal computer with a 2.4 Ghz processor and 4.0 Gb of RAM. The details of simulation results are shown in the following sections.

*5.1. The Impact of Walk on the Performance of SFSOA*

The influence of parameter *Walk* on the performance of the proposed SFSOA approach was examined via three critical values including 0, 0.5 and 1.0 for finding location and size of capacitors in a 33-node distribution network. There are three installation options for the system including (1) one capacitor ($Nc = 1$), (2) two capacitors ($Nc = 2$), and (3) three capacitors ($Nc = 3$). In each study case (installation option), the implementation was performed in 50 trial runs for one value of *Walk*. Thus, there are a total of three iterations of 50 trial runs for each study case corresponding to *Walk* = 0, 0.5 and 1, respectively. Two other parameters of SFSOA consisting of *Npo* and $H_{It}$ are constantly set to 10 and 30 for all study cases. The statistical results through 50 trial runs were analyzed as the minimum, mean and maximum power losses. For the case with $Nc = 1$, the statistical results are reported in Figure 5 while the histogram of total power loss of 50 individual runs is plotted in Figure 6. Similarly, for the cases with $Nc = 2$ and $Nc = 3$, the statistical results and the histogram of total power loss are shown in Figure 7, Figure 8, and Figure 9 and Figure 10 respectively.

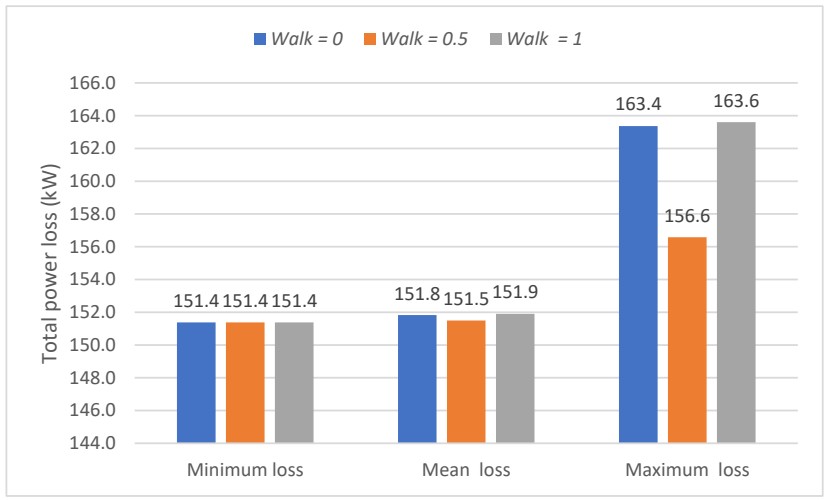

**Figure 5.** Results obtained by setting three values to *Walk* for placing one capacitor in a 33-node distribution network.

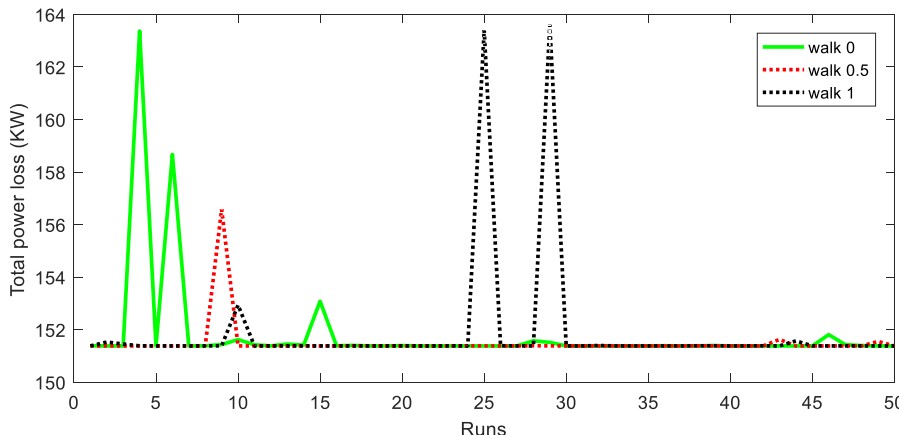

**Figure 6.** Total power loss of 50 runs with three values of *Walk* for placing one capacitor in a 33-node distribution network.

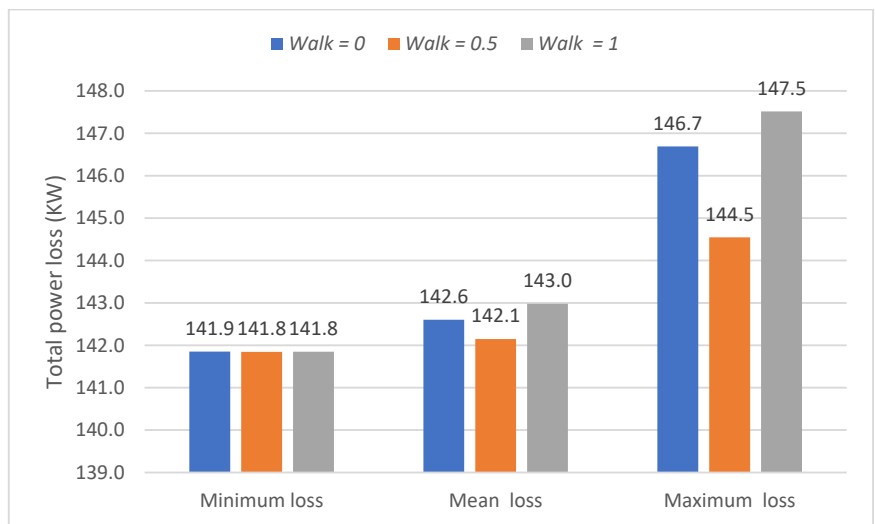

**Figure 7.** Results obtained by setting three values to *Walk* for placing two capacitors in a 33-node distribution network.

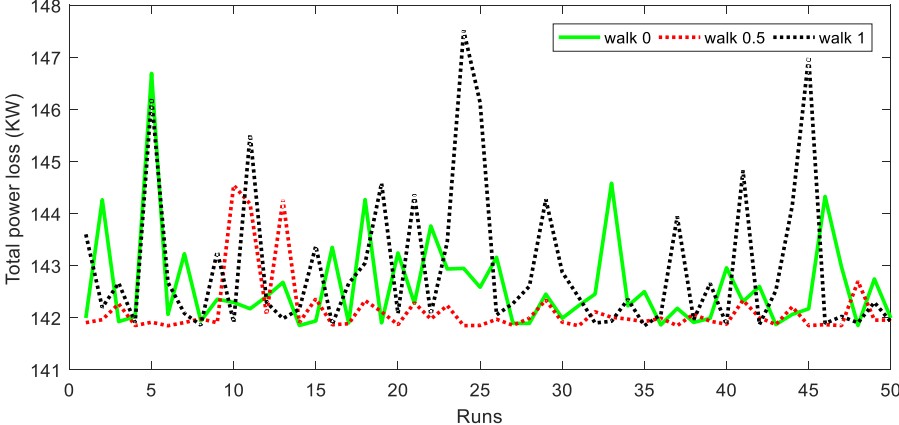

**Figure 8.** Total power loss of 50 runs with three values of *Walk* for placing two capacitors in a 33-node distribution network.

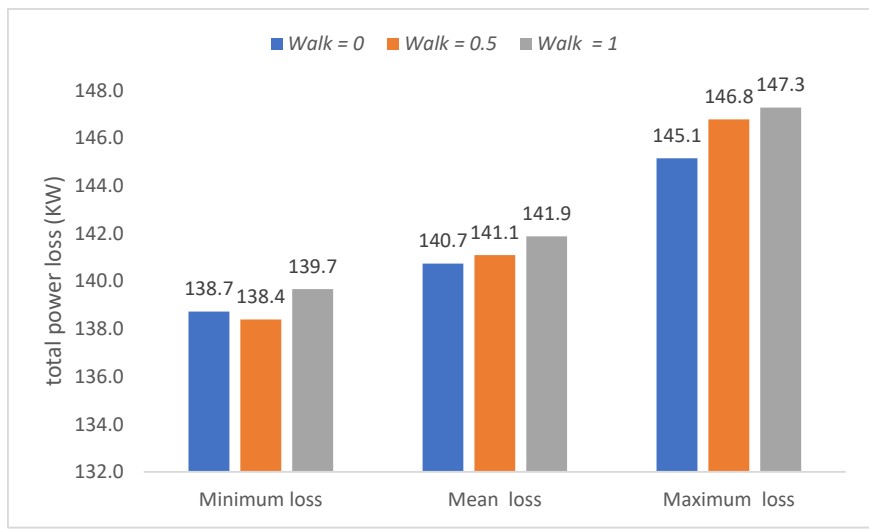

**Figure 9.** Results obtained by setting three values to *Walk* for placing three capacitors in a 33-node distribution network.

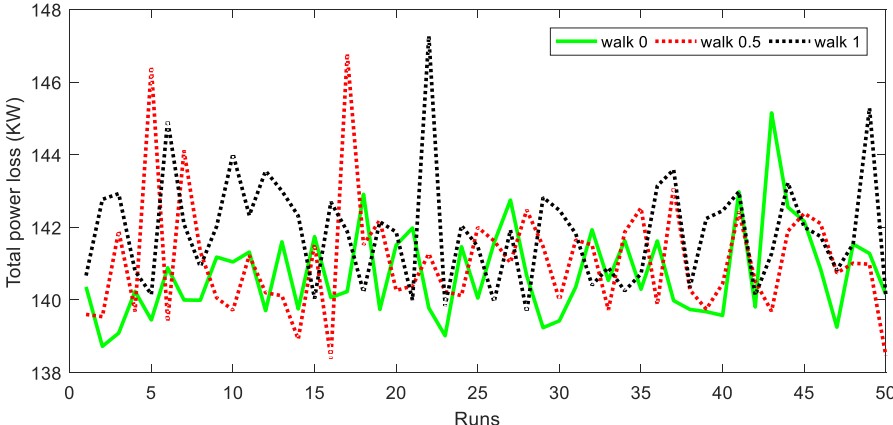

**Figure 10.** Total power loss of 50 runs with three values of *Walk* for placing three capacitors in a 33-node distribution network.

As shown in Figure 5, we could say that *Walk* does not have any impact on the minimum total power loss because SFSOA can reach the same minimum power loss of 151.4 kW for any selected *Walk*. However, it seems to have impacts on the statistical values of SFSOA performance. In fact, *Walk* = 0.5 can improve implementation of SFSOA because it produces the smallest values of the mean and maximum power loss among the three selected *Walk* values. Those statistical values of *Walk* = 0.5 are (151.4, 151.5, 151.6) kW compared to (151.4, 151.8, 163.4) and (151.4, 151.9, 163.6) corresponding to Walk = 0 and Walk = 1.0, respectively. In addition, as seen in Figure 6, optimal solutions of 50 runs are similar to *Walk* = 0.5 and just had one surge at the ninth run (corresponding to the maximum power loss) while those with *Walk* = 0 and *Walk* = 1.0 had higher fluctuations. Thus, *Walk* = 0.5 is the most suitable for SFSOA in the study case of $Nc$ = 1. Similarly, for the case of $Nc$ = 2, as shown in Figure 7, *Walk* = 0.5 reached the best values of all statistical results (the minimum, mean, and maximum power loss) among three selected *Walk* values. Those statistical values of *Walk* = 0.5 are (141.8, 142.1, 144.5) kW compared to (141.9, 142.6, 146.7) kW and (141.8, 143.0, 147.5) kW corresponding to *Walk* = 0 and *Walk* = 1.0, respectively. Moreover, in Figure 8, we can see a mild fluctuation over 50 runs of *Walk* = 0.5 compared to the stronger fluctuations of *Walk* = 0 and *Walk* = 1.0. Hence, *Walk* = 0.5 is also the most suitable for SFSOA in the study case of $Nc$ = 2. Figures 9 and 10 show the results of the final study case of a 33-node radial distribution network with $Nc$ = 3 where we can see a different trend from the

two above cases with $Nc = 1$ and $Nc = 2$ since $Walk = 0.5$ can reach the best minimum power loss but not the best of the mean and maximum power losses. Those statistical values of $Walk = 0.5$ are (138.4, 141.1, 146.8) kW compared to (138.7, 140.7, 145.1) kW and (139.7, 141.9, 147.3) kW corresponding to $Walk = 0$ and $Walk = 1.0$, respectively. Note that a higher $Nc$ increases several control variables of SFSOA; or, we can say that it increases the complexity of the consideration problem. In all three study cases, $Walk = 0.5$ always produced the best minimum loss. However, it seems that the more the control variables (or higher $Nc$) is, the lower the stability of SF SOA implementation with $Walk = 0.5$ gets. Because the most important issue is the minimum loss rather than the mean loss and the maximum loss, the effectiveness of selecting $Walk = 0.5$ in all three study cases was approved. Consequently, we should select $Walk = 0.5$ in the implementation of SFSOA for 33-node distribution networks.

In the case of placing capacitors and a PV system in the 33-node distribution network, the results obtained by SFSOA for the case of installing one PV system with 20% total active power of all loads ($P_{Load}$) and with less than $P_{Load}$ are reported in Table 2. The results presented in the table employed the best location and the best power of capacitors for the case $Walk = 0.5$ above. The table shows the positive impact of the PV system in reducing power loss of the whole system. The PV system can reduce power loss; however, the number of capacitors is also a significant factor for power loss reduction. For the case of $P_{PV} = 20\% \ P_{Load}$, the loss was reduced to 83.531 kW, 75.383 kW and 72.016 kW for the cases $Nc = 1$, 2 and 3, respectively. The power loss before installing the PV system was 151.4 kW, 141.8 kW and 137.4 kW for the cases $Nc = 1$, 2 and 3, respectively. Similarly, the loss reduction was much higher for the case of $P_{PV} < P_{Load}$ and the loss was 58.458 kW, 50.373 kW and 47.232 kW for the case of $Nc = 1$, 2 and 3, respectively. The mean loss and maximum loss were reduced similarly to the minimum loss for the case $P_{PV} = 20\% \ P_{Load}$ and $P_{PV} < P_{Load}$. In summary, the PV system contributes significantly to the loss reduction of the power system and the capacity of the system is proportional to loss reduction. The higher the capacity of the PV system is, the more significant the loss reduction. In addition, the number of the capacitors also plays a huge role in reduction of loss when installing both capacitors and a PV system.

**Table 2.** Results obtained by installing one PV system after installing capacitors for a 33-node distribution network.

| Study Case | $P_{PV} = 20\% \ P_{Load}$ | | | $P_{PV} < P_{Load}$ | | |
|---|---|---|---|---|---|---|
| | $Nc = 1$ | $Nc = 2$ | $Nc = 3$ | $Nc = 1$ | $Nc = 2$ | $Nc = 3$ |
| Min. loss (kW) | 83.531 | 75.383 | 72.016 | 58.458 | 50.373 | 47.232 |
| Mean loss (kW) | 83.807 | 75.404 | 72.037 | 58.593 | 50.413 | 47.429 |
| Max. loss (kW) | 85.129 | 75.600 | 72.212 | 60.242 | 52.121 | 48.974 |
| Std. dev. | 0.335 | 0.047 | 0.044 | 0.442 | 0.248 | 0.532 |

For a 69-node distribution system, the $Walk$ factor was also set to three values including 0, 0.5 and 1.0 for reaching optimal solutions. As a result, $Walk = 0.5$ can reach the best results for the three cases of installing capacitors in the system and the minimum loss, mean loss, maximum loss and standard deviation for the three cases of $Nc = 1$, 2 and 3 are reported in Table 2. The minimum loss is 152.04 kW, 146.44 kW and 145.12 kW for the cases $Nc = 1$, 2 and 3, respectively. Clearly, the loss reduction is the most effective since the number of capacitors is 3 and all values of the case of $Nc = 3$ are the smallest.

As we continued to install one PV system in the 69-node distribution system, $Walk = 0.5$ continued to be employed for the two study cases with $P_{PV} = 20\% \ P_{Load}$ and $P_{PV} < P_{Load}$. The optimal solutions for installing capacitors shown in Table 3 were employed to install another PV system and the results are given in Table 4. When installing a PV system with 20% of load, the loss was 64.632 kW, 59.345 kW and 58.198 kW for the cases $Nc = 1$, 2 and 3, respectively. The loss was much more reduced when installing a PV system with the capacity less than a full load; the loss was 23.198 kW, 18.144 kW and 17.100 kW for the case of $Nc = 1$, 2 and 3, respectively. The loss for the cases of installing 1, 2 and

3 capacitors shown in Table 3 was 152.04 kW, 146.44 kW and 145.12 kW, respectively. It is clear that installing both capacitors and the PV system can reduce loss much more effectively and higher capacity of PV system can reach better loss.

**Table 3.** Results obtained by installing capacitors in a 69-node distribution network.

| Study Case | $Nc = 1$ | $Nc = 2$ | $Nc = 3$ |
|---|---|---|---|
| Min. loss (kW) | 152.04 | 146.44 | 145.12 |
| Mean loss (kW) | 154.70 | 146.60 | 145.49 |
| Max. loss (kW) | 199.53 | 149.27 | 146.61 |
| Std. dev. | 9.49 | 0.576 | 0.477 |

**Table 4.** Results obtained by installing one PV system after installing capacitors for a 69-node distribution network.

| Study Case | $P_{PV} = 20\% \ P_{Load}$ | | | $P_{PV} < P_{Load}$ | | |
|---|---|---|---|---|---|---|
| | $Nc = 1$ | $Nc = 2$ | $Nc=3$ | $Nc = 1$ | $Nc = 2$ | $Nc = 3$ |
| Min. loss (kW) | 64.632 | 59.345 | 58.198 | 23.198 | 18.144 | 17.100 |
| Mean loss (kW) | 67.035 | 61.812 | 59.399 | 26.413 | 24.511 | 23.457 |
| Max. loss (kW) | 123.833 | 118.462 | 117.224 | 102.92 | 97.733 | 96.549 |
| Std. dev. | 11.712 | 11.683 | 8.345 | 15.777 | 21.811 | 21.773 |

*5.2. The Performance of SFSOA Compared to Other Similar Approaches for the 33-Node Distribution System*

In this section, the best results of SFSOA collected from Section 5.1 are compared to those of other methods shown in Table 5 with the same capacitor installation options. As shown in Table 5, the total power loss of the bare network without capacitor placement was 211 kW. When installing one capacitor ($Nc = 1$), the power loss was reduced by applying NFBHA [32] and SFSOA, which remained at about 151.55 kW and 151.37 kW, respectively. When installing two capacitors ($Nc = 2$), both NFBHA [32] and SFSOA continued to reduce the loss to 141.9 kW and 141.84 kW, respectively. Thus, SFSOA is more effective than NFBHA because the of the greater saving powers, 0.18 kW and 0.06 kW corresponding to $Nc = 1$ and $Nc = 2$, respectively. Installing three capacitors ($Nc = 3$) led to lower power loss compared to those of the bare system and two previous compensation options ($Nc = 1$ and $Nc = 2$) where SFSOA was more effective than other methods thanks to the best minimum power loss of 138.41 kW. Furthermore, SFSOA had a small computational burden with the population size of 10 and maximum iterations of 30 while other methods did not report these factors, excluding BFOA [24] with 50 for both population and iterations. The lower population and the smaller number of iterations shows the more promising search ability of an executed method [38].

**Table 5.** Comparisons for the 33-node distribution system.

| Study Case | Method | $Npo$ | $H_{It}$ | Bus (Size) (kVAr) | Total Capacity (kVAr) | Power Loss (KW) |
|---|---|---|---|---|---|---|
| $Nc = 0$ | - | - | - | - | | 211 |
| $Nc = 1$ | NFBHA [32] | - | - | 30 (1190) | 1190 | 151.55 |
| | SFSOA | 10 | 30 | 30 (1258) | 1258 | 151.37 |
| $Nc = 2$ | NFBHA [32] | - | - | 13 (405), 30 (1052) | 1457 | 141.9 |
| | SFSOA | 10 | 30 | 12 (473), 30 (1059) | 1522 | 141.84 |

**Table 5.** *Cont.*

| Study Case | Method | *Npo* | $H_{It}$ | Bus (Size) (kVAr) | Total Capacity (kVAr) | Power Loss (KW) |
|---|---|---|---|---|---|---|
| | BFOA [24] | 50 | 50 | 18(349.6), 30 (820.6), 33 (277.3) | 1447.5 | 144.04 |
| | FPA [27] | NR | NR | 13 (450), 24 (450), 30 (900) | 1800 | 139.075 |
| *Nc* = 3 | PLSF-AA [31] | - | - | 13 (359), 24 (520), 30 (1016) | 1895 | 138.37 |
| | NFBHA [32] | - | - | 13 (383), 25 (386), 30 1000) | 1769 | 138.65 |
| | SFSOA | 10 | 30 | 14 (335), 24 (539), 30 (1050) | 1924 | 138.41 |

However, if total reactive power of all capacitors is considered a comparison criterion, SFSOA was not the best method because the total reactive power of all capacitors was the highest value among compared methods. However, in light of the benefits, including reduction of power loss and satisfaction of the constraint of total installed reactive power of capacitors, SFSOA is still better than other comparable methods.

### 5.3. The Performance of SFSOA Compared to Other Similar Approaches for the 69-Node Distribution Network

A more complicated configuration with the 69-node distribution network was simulated to investigate the performance of SFSOA. The experiments included three installation options, *Nc* = 1, *Nc* = 2 and *Nc* = 3, implemented by using three fixed factors: *Walk*, *Npo*, and $H_{It}$, which were respectively set to 0.5, 10 and 40. The best results of SFSOA were compared to those of other ones shown in Table 6.

**Table 6.** Comparisons for the 69-node distribution system.

| Study Case | Method | *Npo* | $H_{It}$ | Bus (Size) (kVAr) | Total Capacity (KVAr) | Total Loss (KW) |
|---|---|---|---|---|---|---|
| *Nc* = 0 | - | - | - | - | - | 225 |
| *Nc* = 1 | SFSOA | 10 | 40 | 61 (1330) | 1330 | 152.04 |
| *Nc* = 2 | RCGA [20] | 30 | 1000 | 61 (1029), 64 (207) | 1236 | 152.0541 |
| | SFSOA | 10 | 40 | 17 (361), 61 (1275) | 1636 | 146.44 |
| | Two-step [13] | - | - | 19 (225), 63 (900), 63 (225) | 1350 | 148.91 |
| | CIF-PSO [15] | NR | NR | 46 (241), 47 (365), 50 (1015) | 1621 | 152.48 |
| | TLA [23] | 50 | 100 | 12 (600), 61 (1050), 64 (150) | 1800 | 146.35 |
| | FPA [27] | NR | NR | 11 (450), 22 (150), 61 (1350) | 1950 | 145.86 |
| *Nc* = 3 | CSA [28] | 50 | NR | 18 (350), 61 (1150), 65 (65) | 1565 | 146.1 |
| | MSA [30] | 50 | 100 | 12 (450), 21 (150), 61 (1200) | 1800 | 145.41 |
| | PLSF-AA [31] | - | - | 11 (368), 21 (231), 61 (1196) | 1795 | 145.21 |
| | SFSOA | 10 | 40 | 11 (412), 21 (230), 61 (1232) | 1874 | 145.11 |

The table shows that total power loss of the bare network, without installing any capacitors, was 225 kW. Meanwhile, other solutions with installing capacitors can reduce the total power loss significantly. Applying SFSOA for optimally determining the location and size of capacitors in the 69-node distribution network can obtain the power loss values of 152.04 KW, 146.44 kW, and 145.11 kW corresponding to $Nc = 1$, $Nc = 2$ and $Nc = 3$, respectively. Obviously, the power loss is significantly decreased when the number of capacitors is increased.

When installing two capacitors ($Nc = 2$), comparing the power loss of SFSOA and RCGA [20] indicates that SFSOA is more effective because the loss of SFSOA was 146.44 kW but that of RCGA was 152.0541 kW. The locations of the two capacitors found by RCGA were nodes 61 and 64 while those found by SFSOA were 17 and 61. Notably, 61 and 65 were in the same main feeder while 17 and 61 were located at different main feeders. Furthermore, SFSOA was much faster than RCGA in finding the location and size of these capacitors because it used $Npo = 10$ and $H_{It} = 40$ but RCGA used $Npo = 30$ and $H_{It} = 1000$.

For the case with $Nc = 3$, other methods found the worst power loss of 152.48 kW and the best power loss of 145.21 kW while the power loss of SFSOA was 145.11 KW, which was lower than the worst loss and the best loss of other ones by 7.37 kW and 0.1 kW, respectively. Regarding the search speed comparison, SFSOA was faster than other ones such as TLA [23], CSA [28] and MSA [30]. These methods were run by setting population to 50 and the number of iterations to 100 excluding CSA without showing the number of iterations. The comparison between the Two-step method [13] and PLSF-AA [31] were not accomplished because these methods are not metaheuristic algorithms but are instead based on configuration of networks. In summary, SFSOA was more powerful than other ones in the case of three capacitors in terms of finding less power loss, using a smaller population and using a smaller number of iterations.

From the analysis of obtained results in different cases of the number of capacitors, we can conclude that the total power loss of distribution networks can be reduced more if the number of capacitors is increased more. For this optimization problem, SFSOA is more effective than other compared algorithms in finding suitable locations and sizes of capacitors to reduce the total power loss.

*5.4. The Impact of Capacitors and PV Systems on the Power Loss Reduction and Votlage Profile Improvement*

In this section, we discuss the quantitation impact of installing capacitors on radial distribution systems in terms of the power loss reduction and voltage profile enhancement. The total reactive power (kVAr), total power loss (TPL) as well as the power loss reduction (PLR) in kW and in % are presented in Table 7. Given the number of capacitors and Total kVAr, it is apparent that the total kVAr increased once the number of capacitors increased. In fact, for the cases $Nc = 1$, $Nc = 2$ and $Nc = 3$, the total compensated capacity was 1258, 1447.5 and 1922 kVAr, respectively, in the 33-node distribution system, and 1330, 1636 and 1874 kVar in the 69-node distribution system. The greater the number of capacitors installed, the more the total compensated capacity needs, and the better the power loss reduction obtained. For example, for the 33-node distribution system, the reductions in kW compared to the bare system were 59.63, 66.96 and 72.59 kW corresponding to $Nc = 1$, $Nc = 2$, and $Nc = 3$, respectively, and the reductions in percent were 28.26%, 31.73%, and 34.40%, respectively. Similarly, for the 69-node distribution system, the reductions in kW were 72.96, 78.56 and 79.89 kW, respectively, and the reductions in percent were 32.43%, 51.67% and 54.55% corresponding to $Nc = 1$, $Nc = 2$, and $Nc = 3$, respectively. Therefore, installing capacitors in distribution systems makes a significant contribution to the reduction of total power loss.

**Table 7.** Analysis of impact of capacitor and PV system placement on TPL.

| Study Case | Number of Capacitors | 33-Node Network | | | | 69-Node Network | | | |
|---|---|---|---|---|---|---|---|---|---|
| | | Total kVAr | TPL (KW) | PLR (kW) | PLR (%) | Total kVAr | TPL (KW) | PLR (kW) | PLR (%) |
| Without PV system | $Nc = 0$ | - | 211 | - | - | - | 225 | - | - |
| | $Nc = 1$ | 1258 | 151.37 | 59.63 | 28.26 | 1330 | 152.04 | 72.96 | 32.43 |
| | $Nc = 2$ | 1447.5 | 144.04 | 66.96 | 31.73 | 1636 | 146.44 | 78.56 | 34.92 |
| | $Nc = 3$ | 1922 | 138.41 | 72.59 | 34.4 | 1874 | 145.11 | 79.89 | 35.51 |
| With PV system ($20\% P_{Load}$) | $Nc = 1$ | 1258 | 83.531 | 127.469 | 60.41 | 1330 | 64.632 | 160.368 | 71.27 |
| | $Nc = 2$ | 1447.5 | 75.383 | 135.617 | 64.27 | 1636 | 59.345 | 165.655 | 73.62 |
| | $Nc = 3$ | 1922 | 72.016 | 138.984 | 65.87 | 1874 | 58.198 | 166.802 | 74.13 |
| With PV system ($<P_{Load}$) | $Nc = 1$ | 1258 | 58.458 | 152.542 | 72.29 | 1330 | 23.198 | 201.802 | 89.69 |
| | $Nc = 2$ | 1447.5 | 50.373 | 160.627 | 76.13 | 1636 | 18.144 | 206.856 | 91.94 |
| | $N c= 3$ | 1922 | 47.232 | 163.768 | 77.62 | 1874 | 17.1 | 207.9 | 92.4 |

For the case of capacitor and PV system placement, the power loss reduction was more effective. For the case with $P_{PV} = 20\% P_{Load}$, the power loss reduction was 127.469, 135.617 and 138.984 kW corresponding to the reduction of 60.41%, 64.27% and 65.87% for the 33-node distribution system and the power loss reduction was 160.368, 165.655 and 166.802 kW corresponding to the reduction of 71.27%, 73.62% and 74.13% for the 69-node distribution network. As capacity of PV system increased to lower than the full active power of loads, the loss reduction was much more significant. Namely, the power loss reduction was 152.542, 160.627 and 163.768 kW corresponding to 72.29%, 76.13% and 77.62% for the 33-node distribution network and the power loss reduction was 201.802, 206.856, 207.9 kW corresponding to 89.69%, 91.94% and 92.40% for the 69-node distribution network. Thus, the combination of capacitors and PV systems can reduce power loss significantly, especially for the case with high PV system capacity.

To analyze another benefit of installing capacitors and PV systems in distribution networks, the voltage profile according to different installation options is plotted in Figures 11–20. In these figures, we have only considered the PV system with 20% load. Figures 11 and 12 show the voltage profile of the 33 and 69-node distribution systems for the case with capacitor placement, respectively. Figures 11 and 12 also show that installing one capacitor ($Nc = 1$) can improve the voltage drops significantly as compared to the base system without capacitor placement and that notable improvement can be seen when installing two capacitors compared to the case $Nc = 1$. However, it seems that installing three capacitors is not superior to installing just two capacitors where some nodes have better voltage improvements due to $Nc = 2$ while others have better improvements due to $Nc = 3$. Figures 13 and 14 show the voltage profile of the two systems with capacitors and one PV system placement. The two figures are different from Figures 11 and 12 since the voltage profile of the case with one capacitor and one PV system is much better than the voltage profile of the case with only one capacitor. However, the voltage improvement of the case with two capacitors and one PV system was not much better than that of the case with one capacitor and one PV system. Similarly, voltage improvement of the case with three capacitors and one PV system was not much better than that of the case with one capacitor and one PV system, and the case with two capacitors and one PV system.

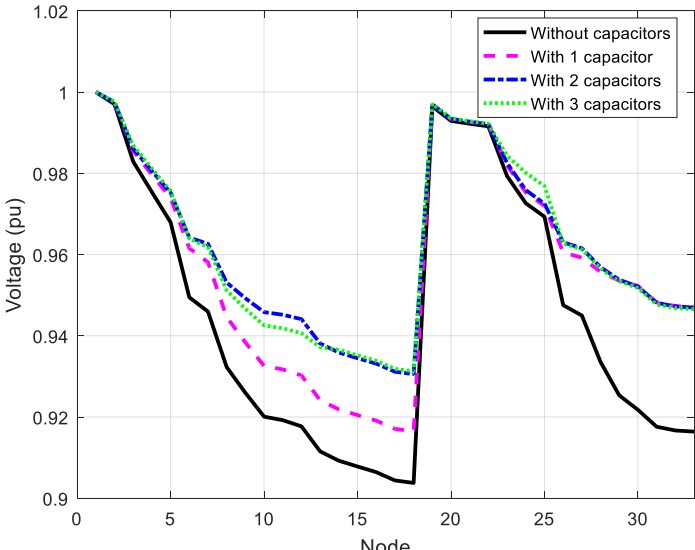

**Figure 11.** Voltage of nodes in the 33-node system for different cases of capacitor placement.

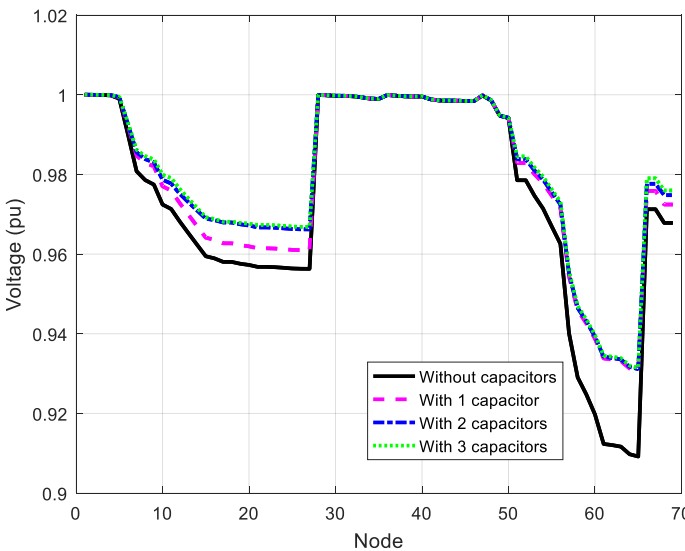

**Figure 12.** Voltage of nodes in the 69-node system for different cases of capacitor placement.

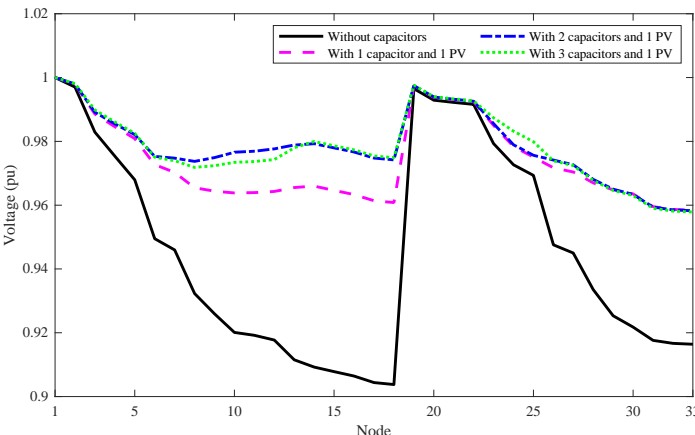

**Figure 13.** The impact of capacitors and PV systems on the voltage profile improvement for the 33-node distribution network.

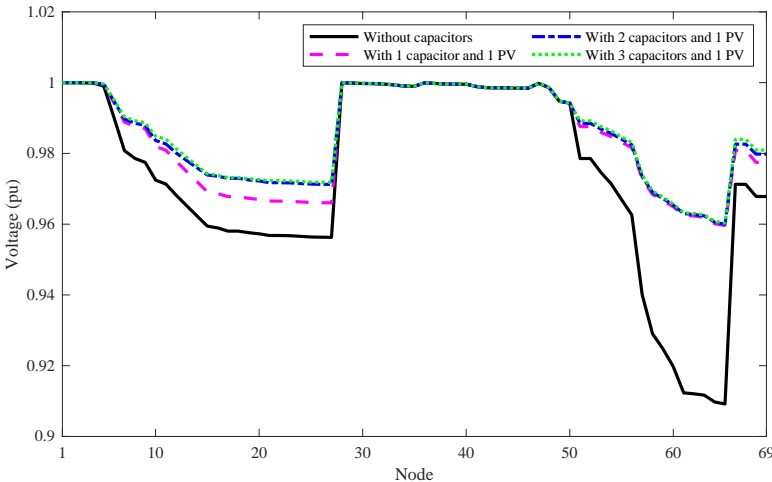

**Figure 14.** The impact of capacitors and PV systems on the voltage profile improvement for the 69-node distribution network.

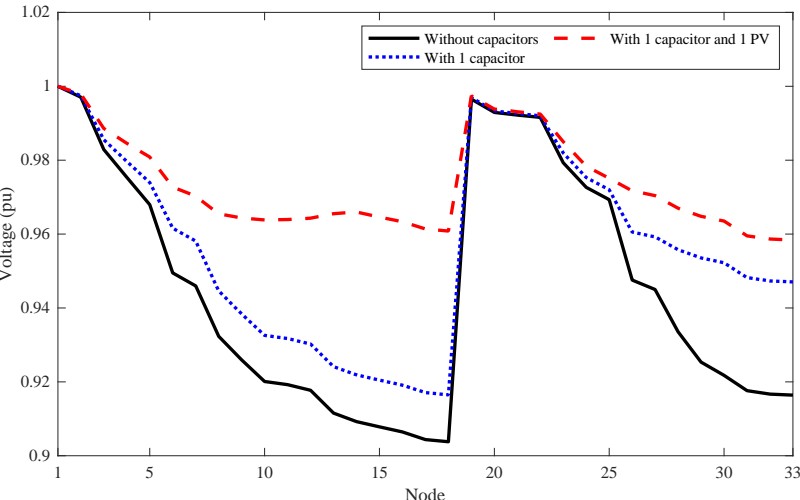

**Figure 15.** The voltage profile improvement for the case of one capacitor and one PV system for the 33-node distribution network.

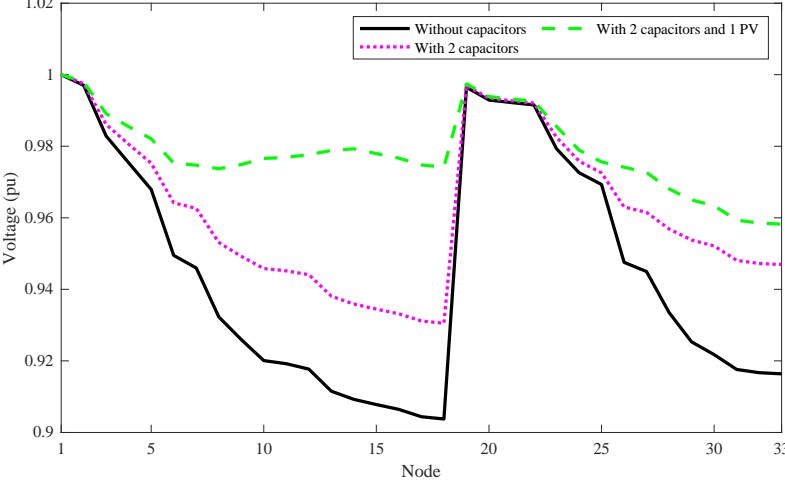

**Figure 16.** The voltage profile improvement for the case of two capacitors and one PV system for the 33-node distribution network.

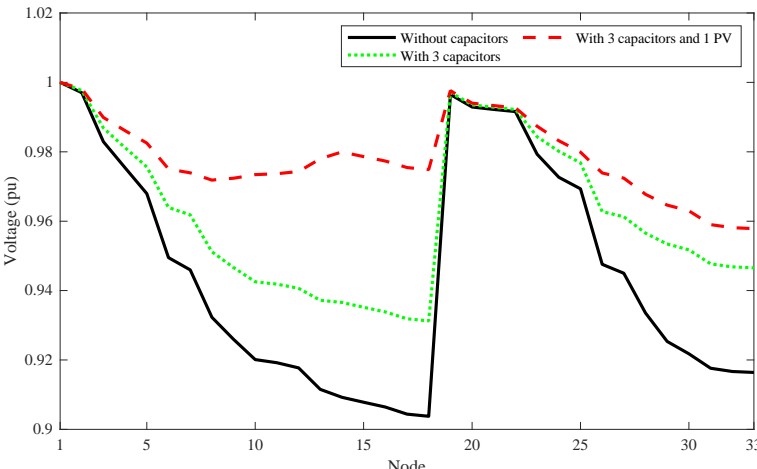

**Figure 17.** The voltage profile improvement for the case of three capacitors and one PV system for the 33-node distribution network.

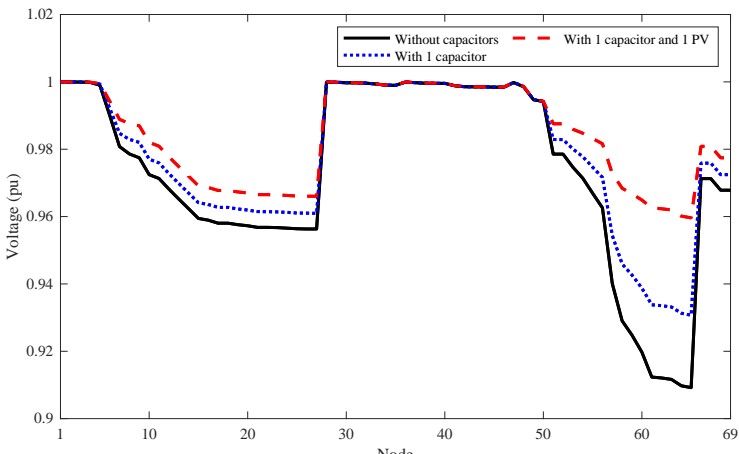

**Figure 18.** The voltage profile improvement for the case of one capacitor and one PV system for the 69-node distribution network.

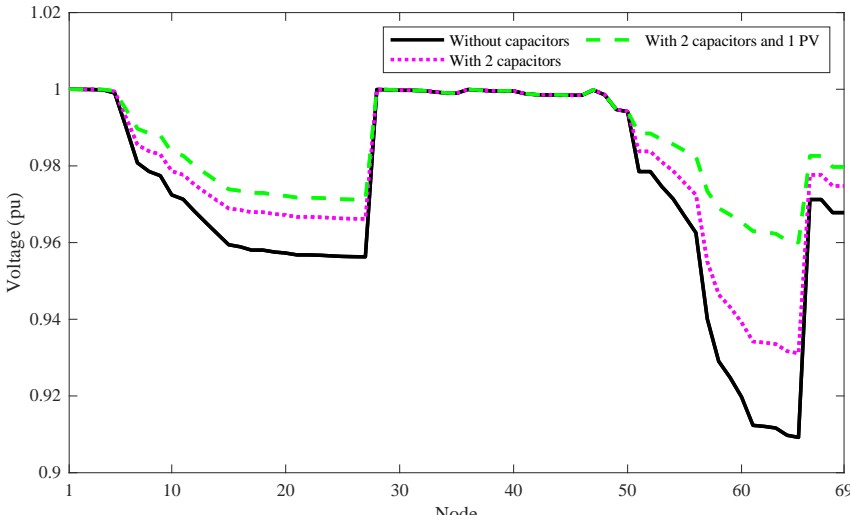

**Figure 19.** The voltage profile improvement for the case of two capacitors and one PV system for the 69-node distribution network.

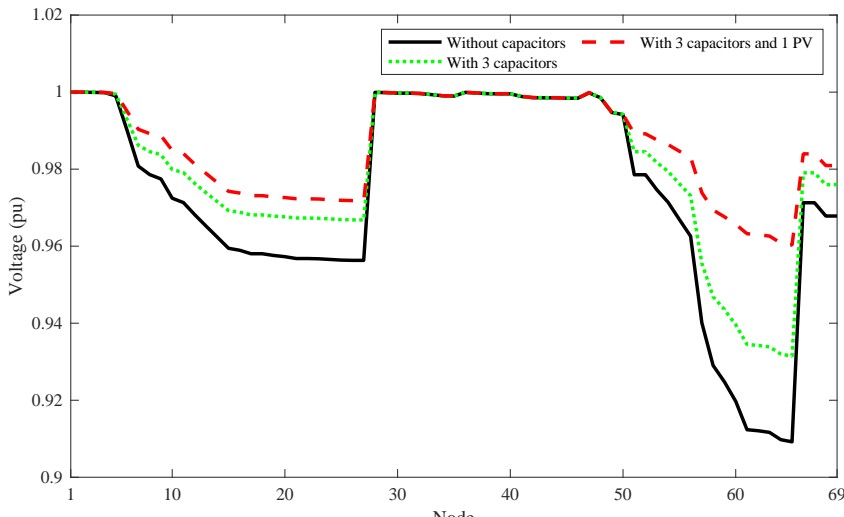

**Figure 20.** The voltage profile improvement for the case of three capacitors and one PV system for the 69-node distribution network.

In Figures 11 and 12, the lowest voltage of the base network was about 0.9 and 0.91 Pu while the lowest voltage of the network with capacitor placement was about 0.93 Pu. So, the voltage improvement, thanks to the capacitor placement, is 3.3% and 2.2% for the 33 and 69-node networks, respectively. As shown in Figures 13 and 14, the lowest voltage of the networks with capacitor and PV system placement was 0.96 Pu. The better voltage is equivalent to the improvements of 6.7% and 5.5%. The placement of capacitors and PV systems can enhance voltage profile of distribution networks significantly.

In order to distinguish the difference between networks with both capacitor and PV system and those with only capacitor placement, Figures 15–20 have been plotted. Figure 15 shows the comparison of voltage profile for the case with one capacitor and the case with one capacitor and one PV system in the 33-node distribution network. The voltage profile shows that the lowest voltage of the case with one capacitor was under 0.92 Pu while that of the case with one capacitor and one PV system was higher than 0.96. The voltage deviation is 0.04 and corresponds to a voltage improvement of 4.4%. Figure 16 shows the comparison of voltage profiles for the case with two capacitors and the case with two capacitors and one PV system in the 33-node distribution network. The lowest voltage of the case with two capacitors was under 0.94 Pu while that of the case with two capacitors and one PV system was higher than 0.96 Pu. The difference is equivalent to the voltage improvement of 2.1%. The calculation for Figure 17 has the same result as Figure 16. Similarly, the voltage improvement for the 69-node distribution network for the case with capacitors and PV systems placement was more significant than the case with only capacitor placement. Figures 18–20 indicate the lowest voltage of the case with only capacitor placement was about 0.93 whereas that of the case with capacitors and PV systems placement was higher than 0.96 Pu. The voltage improvement was equivalent to 3.2%. As a result, the impact of capacitors and PV systems placement can be stated as follows:

(1) Higher number of capacitors require higher total compensated capacity;
(2) Power loss decreases once the total installed capacity increases;
(3) Both capacitor and PV system placement can reach higher power loss reduction and better voltage profile; and
(4) Voltage profile is improved significantly when installing one or two capacitors in the test distribution systems; however, the improvement is not in direct proportion to the compensated capacity.

*5.5. Discussion on the Capacitor and PV System Placement*

5.5.1. Discussion on the Objective Function of Loss Reduction

In this study, we have demonstrated the huge contribution of capacitors and PV system placement to power loss reduction and voltage profile improvement, while other factors regarding economic issues have not been considered. The fact that the investment in capacitors and PV systems is an economic issue is highly considered. As we have stated in the introduction, previous studies have focused on the objective of reducing power loss in distribution lines but have not considered other economic issues such as price of electricity, capacitor cost (including capacitor installation cost and capacitor operation cost) and PV cost (including PV installation cost and PV operation cost). If all costs regarding capacitors and PV systems can be exactly obtained, the problem of placing capacitors and PV system becomes more practical and more useful in distribution systems. A real objective function can be formulated as follows:

$$\text{Maximize } Benefit = Price_e \times En_{re} - Cost_{Cap} - Cost_{PV} \tag{26}$$

where $Price_e$ is the electricity price; $En_{re}$ is the energy loss reduction; $Cost_{Cap}$ and $Cost_{PV}$ are the costs regarding capacitor placement and PV system placement, respectively.

Hence, the objective function of the problem is to maximize benefit instead of reducing total power loss. If information on capacitor installation cost, capacitor operation cost, PV installation cost and PV operation cost is collected correctly, this new objective function is really useful in distribution systems. However, the problem with maximizing benefit is that it has to cope with difficulties of calculating the operation costs of capacitors and PV systems. Furthermore, PV system installation is also related to ground price and ground area. Hence, it is hard to get information on the exact costs for a useful objective function. For the application of the objective function (13), the assumptions below are considered:

- The installation of capacitors in distribution systems must be accomplished by power companies for the purpose of reducing loss and improving voltage profile.
- The installation of PV systems must be accomplished by power companies due to the requirement of reducing power from thermal power plants for reducing polluted emissions to the air and for increasing renewable energies.

5.5.2. Discussion on the Geography Location Constraint for PV System Placement

In Section 5.1, we implemented PV system placement in distribution networks without any constraints on discovering suitable locations for building up PV power plants where they are directly influenced by ground area, ground price and solar radiation. In the study, we applied IEEE distribution systems with 33 and 69 buses but we do not have information on geographical locations. So, the first two factors such as ground area and ground price can be ignored. Meanwhile, the suitable locations (suitable buses) were not considered in the sections above. Consequently, we have implemented other simulations with the assumption that some buses are suitable for the placement of PV systems. The assumption is as follows:

(1) For the 33-node distribution network, suitable nodes for the PV system placement are 5, 6, 16, 17, 18, 20, 21, 22, 24, 25, 29, 30, 31 and 32.
(2) For the 69-node distribution network, suitable nodes for the PV system placement are 20, 21, 22, 27, 34, 35, 43, 44, 45, 46, 48, 49, 50, 63, 64 and 65.

The results from additional simulations have been added in Tables 8–11 for comparisons. For the 33-node distribution network with the limitation of $P_{PV} = 20\% \, P_{Load}$ shown in Table 8, node 14 was more suitable than node 16 for PV system placement. As the PV system was located at node 14, the power loss was equal to 83.531 kW, 75.383 kW and 72.016 kW for $Nc = 1$, $Nc = 2$ and $Nc = 3$, respectively.

As node 14 was not allowed to place PV systems, node 16, which is a neighboring node, was selected for PV system placement. The power loss caused by installing PV system at node 16 was higher and equal to 85.43 kW, 77.18 kW and 73.806 kW for $Nc = 1$, $Nc = 2$ and $Nc = 3$, respectively. The comparisons indicate that if there is no limitation on geographical location for PV system placement, power loss reduction can be reached more effectively. Conversely, in Table 9 with the limitation of $P_{PV} < P_{Load}$, the most suitable node for PV system placement was six for both cases with and without constraint on geographical location. Coincidentally, the most suitable node (node 6) was not constrained by geography location. As a result, power loss was 58.458 kW for $Nc = 1$, 50.373 kW for $Nc = 2$ and 47.232 kW for $Nc = 3$ for cases with and without constraint of geography location. The results indicate that if the most suitable locations for PV system placement are not eliminated, the distribution networks also reach power loss reduction effectively.

**Table 8.** Results obtained by installing one PV system with $P_{PV} = 20\%\ P_{Load}$ after installing capacitors for 33-node distribution network.

| Study Case | $P_{PV} = 20\%\ P_{Load}$ | | | $P_{PV} = 20\%\ P_{Load}$ & Constrained PV Location | | |
|---|---|---|---|---|---|---|
| | $Nc = 1$ | $Nc = 2$ | $Nc = 3$ | $Nc = 1$ | $Nc = 2$ | $Nc = 3$ |
| Min. loss (kW) | 83.531 | 75.383 | 72.016 | 85.43 | 77.18 | 73.806 |
| Mean loss (kW) | 83.807 | 75.404 | 72.037 | 85.72 | 77.18 | 73.929 |
| Max. loss (kW) | 85.129 | 75.600 | 72.212 | 91.18 | 77.24 | 79.059 |
| Std. dev. | 0.335 | 0.047 | 0.044 | 1.15 | 0.01 | 0.742 |
| PV location | 14 | 14 | 14 | 16 | 16 | 16 |
| Size of PV (kW) | 742.97 | 743 | 143 | 143 | 743 | 743 |

**Table 9.** Results obtained by installing one PV system with $P_{PV} < P_{Load}$ after installing capacitors for 33-node distribution network.

| Study Case | $P_{PV} < P_{Load}$ | | | $P_{PV} < P_{Load}$ & Constrained PV Location | | |
|---|---|---|---|---|---|---|
| | $Nc = 1$ | $Nc = 2$ | $Nc = 3$ | $Nc = 1$ | $Nc = 2$ | $Nc = 3$ |
| Min. loss (kW) | 58.458 | 50.373 | 47.232 | 58.458 | 50.373 | 47.232 |
| Mean loss (kW) | 58.593 | 50.413 | 47.429 | 61.050 | 51.292 | 48.916 |
| Max. loss (kW) | 60.242 | 52.121 | 48.974 | 81.413 | 72.844 | 69.664 |
| Std. dev. | 0.442 | 0.248 | 0.532 | 7.052 | 4.446 | 5.571 |
| PV location | 6 | 6 | 6 | 6 | 6 | 6 |
| Size of PV (kW) | 2531 | 2519.32 | 2517.17 | 2532 | 2519.32 | 2517.17 |

**Table 10.** Results obtained by installing one PV system with $P_{PV} = 20\%\ P_{Load}$ after installing capacitors for 69-node distribution network.

| Study Case | $P_{PV} = 20\%\ P_{Load}$ | | | $P_{PV} = 20\%\ P_{Load}$ & Constrained PV Location | | |
|---|---|---|---|---|---|---|
| | $Nc = 1$ | $Nc = 2$ | $Nc = 3$ | $Nc = 1$ | $Nc = 2$ | $Nc = 3$ |
| Min. loss (kW) | 64.632 | 59.345 | 58.198 | 64.845 | 59.557 | 58.410 |
| Mean loss (kW) | 67.035 | 61.812 | 59.399 | 73.462 | 69.397 | 65.824 |
| Max. loss (kW) | 123.833 | 118.462 | 117.224 | 126.052 | 120.670 | 119.449 |
| Std. dev. | 11.712 | 11.683 | 8.345 | 21.435 | 22.605 | 20.013 |
| PV location | 61 | 61 | 61 | 63 | 63 | 63 |
| Size of PV (kW) | 760 | 760 | 760 | 760 | 760 | 760 |

**Table 11.** Results obtained by installing one PV system with $P_{PV} < P_{Load}$ after installing capacitors for 69-node distribution network.

| Study Case | $P_{PV} < P_{Load}$ | | | $P_{PV} < P_{Load}$ & Constrained PV Location | | |
|---|---|---|---|---|---|---|
| | $Nc = 1$ | $Nc = 2$ | $Nc = 3$ | $Nc = 1$ | $Nc = 2$ | $Nc = 3$ |
| Min. loss (kW) | 23.198 | 18.144 | 17.100 | 26.455 | 21.389 | 20.342 |
| Mean loss (kW) | 26.413 | 24.511 | 23.457 | 52.686 | 41.435 | 38.369 |
| Max. loss (kW) | 102.92 | 97.733 | 96.549 | 126.052 | 120.670 | 119.449 |
| Std. dev. | 15.777 | 21.811 | 21.773 | 43.96 | 40.037 | 38.391 |
| PV location | 61 | 61 | 61 | 63 | 63 | 63 |
| Size of PV (kW) | 1830.25 | 1826.99 | 1826.515 | 1769.32 | 1767.45 | 1766.94 |

In contrast to the 33-node distribution network, the PV system location in the 69-node distribution network shown in Tables 10 and 11 is the same for the two cases with $P_{PV} = 20\% \ P_{Load}$ and $P_{PV} < P_{Load}$ since node 61 was the best location for the case without geography location constraint and node 63 was the best location for the case with geography location constraint. For the case with $P_{PV} = 20\% \ P_{Load}$, the PV system at node 61 could reach a loss of 64.632 kW, 59.345 kW and 58.198 kW whereas the PV system at node 63 had to suffer a higher loss equal to 64.845 kW, 59.557 kW and 58.410 kW for $Nc = 1$, $Nc = 2$ and $Nc = 3$, respectively. Similarly, the PV system at node 61 was also more suitable than at node 63 for the case with $P_{PV} < P_{Load}$ and power loss was much smaller when the PV system was located at node 61.

From the discussion on loss and geography location limit, it is clear that loss reduction is more effective for the case without geographical constraint. In practice, if all nodes in distribution systems can possibly place PV systems, the benefit from the loss reduction is significant and the investment in PV systems in distribution is highly feasible.

### 5.5.3. Discussion on the Change of Loads

In this paper, we have added the data of the two studied distribution networks in Appendix A. All loads have different active power and reactive power. The active power and reactive power of loads are considered only for one period and there is no change of the powers during the single period. In fact, for the problem of installing capacitors and PV systems in the distribution networks (discussed in the introduction), all studies have focused on only a single period and the capacitor placement is dependent only the active power and reactive power of the single period. Thus, in this paper we have followed the previous studies in accepting the assumption that the active power and reactive power of loads in the distribution systems are rated powers, which are the highest values of operation values. SFSOA has been employed to find the location and size of capacitors effectively. Active power from the methods for different numbers of capacitors was less than those from other methods. The results mean that the location and size of capacitors that are chosen by the applied SFSOA was highly effective. In the case where loads change into lower active and reactive powers and the location of capacitors does not change, the determination of generation of the capacitors can be accomplished easily. In such cases, the generation of capacitors must be lower than the determined size of capacitors in cases of rated load. Also, the control variables of the problem are only the generation of capacitors. The task of SFSOA becomes to produce simpler generation of capacitors at predetermined nodes, and then the Backward/forward sweep algorithm is run to calculate the current of branches and voltage of nodes. Finally, active power loss of each branch is obtained by using Equation (14) and total active power loss of the distribution network is calculated by using Equation (13). Fitness function (25) is still applied to measure the quality of solutions produced by SFSOA in which the best generation of capacitors is corresponding to the solution with the lowest value of the fitness function.

5.5.4. Discussion on the Compensation Capacity

As shown in the Table 5 comparison of results from the 33-node distribution network and in the Table 6 comparison of results from the 69-node distribution network, the total compensation from SFSOA was higher than that of other methods. A hasty observation of the two tables could lead to the conclusion that larger capacitors can reduce total power loss more effectively. However, when applying SFSOA for determining the location and size of capacitors, we have used the same constraints as previous studies about the total reactive power of capacitors. All previous studies used the same constraint (17), which is $\sum_{i=1}^{Nc} Q_{ci} \leq Q_c^{\max}$ where $Q_c^{max}$ is equal to the total reactive power of all loads. The total reactive power of loads in the 33 and 69-node distribution networks is 2300 kVAr and 2695 kVAr, respectively. The total compensation obtained by SFSOA for the first system was 1258 kVAr for $Nc = 1$, 1522 kVAr for $Nc = 2$ and 1924 kVAr for $Nc = 3$, and for the second system was 1330 kVAr for $Nc = 1$, 1636 kVAr for $Nc = 2$, and 1874 kVAr for $Nc = 3$ (where $Nc$ is the number of capacitor banks). Clearly, the optimal compensation obtained by SFSOA was less than 2300 kVAr for all cases of the first system and less than 2695 kVAr for all cases of the second system. However, this does not mean that higher compensation capacity can result in less power loss. For example, for the 69-node distribution network with $Nc = 3$, the total compensation capacity of FPA [27] was 1950 kVAr, which was higher than the 1924 kVAr of SFSOA, but the loss of FPA [27] was 145.86 kW, which was higher than the 138.41 kW of SFSOA. Similarly, the two-step method [13] used a total compensation capacity of 1350 kW but its loss was 148.91 kW, which was less than 152.48 kW from CIF-PSO [15], which used the total compensation of 1621 kW. The comparison between TLA [23] and CSA [28] is similar since TLA [23] used the compensation of 1800 kVar but its loss was 146.35 kW while the compensation of CSA [28] was 1565 kVAr but its loss was 146.1 kW. In summary, the problem does not consider the compensation capacity as long as the compensation capacity is not higher than the total reactive power of loads. Therefore, the applied SFSOA method, with less loss than other methods, can be a more effective method of placing capacitors in distribution networks.

## 6. Conclusions

In this study, conventional Stochastic Fractal Search Optimization Algorithm was applied for determining the best size and the most appropriate sitting of capacitors and PV systems in two radial distribution networks including 33 and 69-node systems for increasing the reduction of power loss in distribution lines. Different study cases of the placement of one PV system and different numbers of capacitors including one capacitor, two capacitors and three capacitors for each system have been implemented to find total power losses for performance comparison. In addition, control parameters of the applied method and other comparable methods have been also analyzed for convergence speed comparison. The comparisons show SFSOA was more effective and faster than other ones. The total power losses from SFSOA were either the same or less than those of other but SFSOA always employed a much smaller population size and a smaller number of iterations. Furthermore, the combination of capacitors and the PV system can reduce total power loss and improve voltage profile effectively. In the 33 and 69-node distribution systems, the loss reduction can be up to 65.87% and 74.13% for the case that the PV system's capacity is equal to 20% of load. The reduction is much higher and equal to 77.62% and 92.40% for the case that the PV system's capacity is less than full load. Similarly, the voltage profile can be improved up to 3.3% for only capacitor placement and up to 6.7% for both capacitor and PV system placement. Consequently, we conclude that SFSOA is an efficient method for placing capacitors and PV systems in distribution networks.

**Author Contributions:** T.T.N. (Thuan Thanh Nguyen) and T.T.N. (Thang Trung Nguyen) have implemented the applied method for two distribution networks and written some parts of the article. B.H.D. has edited the whole article. T.D.P. has collected results, made tables and figures. All authors have read and agreed to the published version of the manuscript.

**Funding:** There is no external funding for the work.

**Conflicts of Interest:** The authors declare there is no conflict of interests regarding the publication of the article.

## Nomenclature

| | |
|---|---|
| $C_{It}$ | Current computation iteration. |
| $H_{It}$ | The highest number of iterations |
| $Npo$ | Population size |
| $Ndf$ | The number of diffused solutions |
| $S_{best}$ | The best solution in the current set of solutions |
| $Rd_x$ | A random number of the $x$th solution produced in range of 0 and 1 |
| $S_{rd1}$, $S_{rd2}$ | Randomly chosen solutions from the population |
| $\varepsilon$ | Random number within 0 and 1 |
| $Nc$ | The number of installed capacitors in the distribution networks |
| $IF_x$ | The impact factor of the $x$th solution |
| $Nn$ | The number of nodes in distribution systems |
| $I_l^{max}$ | The possible maximum current of the $l$th distribution line |
| $Nl$ | Number of distribution lines in distribution systems |
| $F_x$ | Fitness function of the $x$th solution |
| $\omega_1$ and $\omega_2$ | Penalty parameters |
| $I_{l,x}$ | Current magnitude in the $l$th branch corresponding to the $x$th solution |
| $U_{m,x}$ | Voltage magnitude of the $m$th node corresponding to the $x$th solution |

## Appendix A

**Table A1.** Data of the 33-node distribution net.

| Branch Number | Sending Node | Receiving Node | Resistance ($\Omega$) | Reactance ($\Omega$) | Nominal Load at Receiving Node | | Maximum Line Capacity (kVA) |
|---|---|---|---|---|---|---|---|
| | | | | | P(kW) | Q(kVAr) | |
| 1 | 1 | 2 | 0.0922 | 0.047 | 100 | 60 | 400 |
| 2 | 2 | 3 | 0.493 | 0.251 | 90 | 40 | 400 |
| 3 | 3 | 4 | 0.3661 | 0.1864 | 120 | 80 | 400 |
| 4 | 4 | 5 | 0.3811 | 0.1941 | 60 | 30 | 400 |
| 5 | 5 | 6 | 0.819 | 0.707 | 60 | 20 | 400 |
| 6 | 6 | 7 | 0.1872 | 0.6188 | 200 | 100 | 300 |
| 7 | 7 | 8 | 1.7117 | 1.2357 | 200 | 100 | 300 |
| 8 | 8 | 9 | 1.0299 | 0.74 | 60 | 20 | 200 |
| 9 | 9 | 10 | 1.044 | 0.74 | 60 | 20 | 200 |
| 10 | 10 | 11 | 0.1967 | 0.0651 | 45 | 30 | 200 |
| 11 | 11 | 12 | 0.3744 | 0.1237 | 60 | 35 | 200 |
| 12 | 12 | 13 | 1.468 | 1.1549 | 60 | 35 | 200 |
| 13 | 13 | 14 | 0.5416 | 0.7129 | 120 | 80 | 200 |
| 14 | 14 | 15 | 0.5909 | 0.526 | 60 | 10 | 200 |
| 15 | 15 | 16 | 0.7462 | 0.5449 | 60 | 20 | 200 |
| 16 | 16 | 17 | 1.2889 | 1.721 | 60 | 20 | 200 |
| 17 | 17 | 18 | 0.732 | 0.5739 | 90 | 40 | 200 |
| 18 | 2 | 19 | 0.164 | 0.1565 | 90 | 40 | 200 |
| 19 | 19 | 20 | 1.5042 | 1.3555 | 90 | 40 | 200 |
| 20 | 20 | 21 | 0.4095 | 0.4784 | 90 | 40 | 200 |
| 21 | 21 | 22 | 0.7089 | 0.9373 | 90 | 40 | 200 |
| 22 | 3 | 23 | 0.4512 | 0.3084 | 90 | 50 | 200 |
| 23 | 23 | 24 | 0.898 | 0.7091 | 420 | 200 | 200 |
| 24 | 24 | 25 | 0.8959 | 0.701 | 420 | 200 | 200 |
| 25 | 6 | 26 | 0.2031 | 0.1034 | 60 | 25 | 300 |
| 26 | 26 | 27 | 0.2842 | 0.1447 | 60 | 25 | 300 |

**Table A1.** *Cont.*

| Branch Number | Sending Node | Receiving Node | Resistance (Ω) | Reactance (Ω) | Nominal Load at Receiving Node | | Maximum Line Capacity (kVA) |
|---|---|---|---|---|---|---|---|
| | | | | | P(kW) | Q(kVAr) | |
| 27 | 27 | 28 | 1.0589 | 0.9338 | 60 | 20 | 300 |
| 28 | 28 | 29 | 0.8043 | 0.7006 | 120 | 70 | 200 |
| 29 | 29 | 30 | 0.5074 | 0.2585 | 200 | 600 | 200 |
| 30 | 30 | 31 | 0.9745 | 0.9629 | 150 | 70 | 200 |
| 31 | 31 | 32 | 0.3105 | 0.3619 | 210 | 100 | 200 |
| 32 | 32 | 33 | 0.3411 | 0.5302 | 60 | 40 | 200 |

**Table A2.** Data of the 69-node distribution network.

| Branch Number | Sending Node | Receiving Node | Resistance (Ω) | Reactance (Ω) | Nominal Load at Receiving Node | | Maximum Line Capacity (kVA) |
|---|---|---|---|---|---|---|---|
| | | | | | P(kW) | Q(kVAr) | |
| 1 | 1 | 2 | 0.0005 | 0.0012 | 0 | 0 | 10,761 |
| 2 | 2 | 3 | 0.0005 | 0.0012 | 0 | 0 | 10,761 |
| 3 | 3 | 4 | 0.0015 | 0.0036 | 0 | 0 | 10,761 |
| 4 | 4 | 5 | 0.0251 | 0.0294 | 0 | 0 | 5823 |
| 5 | 5 | 6 | 0.366 | 0.1864 | 2.6 | 2.2 | 1899 |
| 6 | 6 | 7 | 0.3811 | 0.1941 | 40.4 | 30 | 1899 |
| 7 | 7 | 8 | 0.0922 | 0.047 | 75 | 54 | 1899 |
| 8 | 8 | 9 | 0.0493 | 0.0251 | 30 | 22 | 1899 |
| 9 | 9 | 10 | 0.819 | 0.2707 | 28 | 19 | 1455 |
| 10 | 10 | 11 | 0.1872 | 0.0691 | 145 | 104 | 1455 |
| 11 | 11 | 12 | 0.7114 | 0.2351 | 145 | 104 | 1455 |
| 12 | 12 | 13 | 1.03 | 0.34 | 8 | 5.5 | 1455 |
| 13 | 13 | 14 | 1.044 | 0.345 | 8 | 5.5 | 1455 |
| 14 | 14 | 15 | 1.058 | 0.3496 | 0 | 0 | 1455 |
| 15 | 15 | 16 | 0.1966 | 0.065 | 45.5 | 30 | 1455 |
| 16 | 16 | 17 | 0.3744 | 0.1238 | 60 | 35 | 1455 |
| 17 | 17 | 18 | 0.0047 | 0.0016 | 60 | 35 | 2200 |
| 18 | 18 | 19 | 0.3276 | 0.1083 | 0 | 0 | 1455 |
| 19 | 19 | 20 | 0.2106 | 0.069 | 1 | 0.6 | 1455 |
| 20 | 20 | 21 | 0.3416 | 0.1129 | 114 | 81 | 1455 |
| 21 | 21 | 22 | 0.014 | 0.0046 | 5.3 | 3.5 | 1455 |
| 22 | 22 | 23 | 0.1591 | 0.0526 | 0 | 0 | 1455 |
| 23 | 23 | 24 | 0.3463 | 0.1145 | 28 | 20 | 1455 |
| 24 | 24 | 25 | 0.7488 | 0.2745 | 0 | 0 | 1455 |
| 25 | 25 | 26 | 0.3089 | 0.1021 | 14 | 10 | 1455 |
| 26 | 26 | 27 | 0.1732 | 0.0572 | 14 | 10 | 1455 |
| 27 | 3 | 28 | 0.0044 | 0.0108 | 26 | 18.6 | 10,761 |
| 28 | 28 | 29 | 0.064 | 0.1565 | 26 | 18.6 | 10,761 |
| 29 | 29 | 30 | 0.3978 | 0.1315 | 0 | 0 | 1455 |
| 30 | 30 | 31 | 0.0702 | 0.0232 | 0 | 0 | 1455 |
| 31 | 31 | 32 | 0.351 | 0.116 | 0 | 0 | 1455 |
| 32 | 32 | 33 | 0.839 | 0.2816 | 14 | 10 | 2200 |
| 33 | 33 | 34 | 1.708 | 0.5646 | 19.5 | 14 | 1455 |
| 34 | 34 | 35 | 1.474 | 0.4673 | 6 | 4 | 1455 |
| 35 | 3 | 36 | 0.0044 | 0.0108 | 26 | 18.55 | 10,761 |
| 36 | 36 | 37 | 0.064 | 0.1565 | 26 | 18.55 | 10,761 |
| 37 | 37 | 38 | 0.1053 | 0.123 | 0 | 0 | 5823 |
| 38 | 38 | 39 | 0.0304 | 0.0355 | 24 | 17 | 5823 |
| 39 | 39 | 40 | 0.0018 | 0.0021 | 24 | 17 | 5823 |
| 40 | 40 | 41 | 0.7283 | 0.8509 | 1.2 | 1 | 5823 |

**Table A2.** *Cont.*

| Branch Number | Sending Node | Receiving Node | Resistance (Ω) | Reactance (Ω) | Nominal Load at Receiving Node | | Maximum Line Capacity (kVA) |
|---|---|---|---|---|---|---|---|
| | | | | | P(kW) | Q(kVAr) | |
| 41 | 41 | 42 | 0.31 | 0.3623 | 0 | 0 | 5823 |
| 42 | 42 | 43 | 0.041 | 0.0478 | 6 | 4.3 | 5823 |
| 43 | 43 | 44 | 0.0092 | 0.0116 | 0 | 0 | 5823 |
| 44 | 44 | 45 | 0.1089 | 0.1373 | 39.22 | 26.3 | 5823 |
| 45 | 45 | 46 | 0.0009 | 0.0012 | 39.22 | 26.3 | 6709 |
| 46 | 4 | 47 | 0.0034 | 0.0084 | 0 | 0 | 10,761 |
| 47 | 47 | 48 | 0.0851 | 0.2083 | 79 | 56.4 | 10,761 |
| 48 | 48 | 49 | 0.2898 | 0.7091 | 384.7 | 274.5 | 10,761 |
| 49 | 49 | 50 | 0.0822 | 0.2011 | 384 | 274.5 | 10,761 |
| 50 | 8 | 51 | 0.0928 | 0.0473 | 40.5 | 28.3 | 1899 |
| 51 | 51 | 52 | 0.3319 | 0.1114 | 3.6 | 2.7 | 2200 |
| 52 | 9 | 53 | 0.174 | 0.0886 | 4.35 | 3.5 | 1899 |
| 53 | 53 | 54 | 0.203 | 0.1034 | 26.4 | 19 | 1899 |
| 54 | 54 | 55 | 0.2842 | 0.1447 | 24 | 17.2 | 1899 |
| 55 | 55 | 56 | 0.2813 | 0.1433 | 0 | 0 | 1899 |
| 56 | 56 | 57 | 1.59 | 0.5337 | 0 | 0 | 2200 |
| 57 | 57 | 58 | 0.7837 | 0.263 | 0 | 0 | 2200 |
| 58 | 58 | 59 | 0.3042 | 0.1006 | 100 | 72 | 1455 |
| 59 | 59 | 60 | 0.3861 | 0.1172 | 0 | 0 | 1455 |
| 60 | 60 | 61 | 0.5075 | 0.2585 | 1244 | 888 | 1899 |
| 61 | 61 | 62 | 0.0974 | 0.0496 | 32 | 23 | 1899 |
| 62 | 62 | 63 | 0.145 | 0.0738 | 0 | 0 | 1899 |
| 63 | 63 | 64 | 0.7105 | 0.3619 | 227 | 162 | 1899 |
| 64 | 64 | 65 | 1.041 | 0.5302 | 59 | 42 | 1899 |
| 65 | 11 | 66 | 0.2012 | 0.0611 | 18 | 13 | 1455 |
| 66 | 66 | 67 | 0.0047 | 0.0014 | 18 | 13 | 1455 |
| 67 | 12 | 68 | 0.7394 | 0.2444 | 28 | 20 | 1455 |
| 68 | 68 | 69 | 0.0047 | 0.0016 | 28 | 20 | 1455 |

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
