# Peer review of "Active Power Loss Reduction for Radial Distribution Systems by Placing Capacitors and PV Systems with Geography Location Constraints"

_sustainability, doi:10.3390/su12187806_

Round 1

Reviewer 1 Report

In this paper, the authors developed a method to optimize the location of PV and capacitors in the distribution system, in order to minimize the power loss and voltage drop. SFSOA method is used to search the optimization solution.

The major concern of this paper is whether it is practical. The installation of capacitor banks and PV could be a big investment, considering the length of the distribution grid comparing with the transmission. The benefit brought by this method needs to be compared with the investment, to prove its feasibility. E.g., the investment could be included in the objective function.

Furthermore, the location of PV depends on a lot of factors, such as the feasibility of the ground, the ground price, and solar radiation. Therefore, the location selection of PV solely based on the power loss reductions is implausible.

The loads considered in this paper are all consistent, and they are the same at each node. In reality, they change a lot. How to choose the size of the capacitors regarding this change is not considered in this paper.

When comparing with other algorithms, it appears that the larger capacitor is used, the more power loss is reduced, which sounds straightforward. How does it prove the effectiveness of the method proposed in this paper?

Other comments:

Line 131-132, the description of nodes does not match Figure 1.

Equation (1) does not look correct to me. The voltage loss should be

(see attachment for the equation)

Author Response

Thank you very much for your comments and suggestions. Please see the attachment for our detailed response.

Reviewer 2 Report

This is a very interesting paper. My suggestions would be:

  • To put the summary of literature in Section 1 into a Table format
  • Can Tables 4&5 be transform in a Table format to ease reader to compare the power losses? Or Maybe add new figure just to show this? As it is, the Tables are quite long.
  •  

Author Response

(The authors gave the same response as above.)

Reviewer 3 Report

The manuscript submitted to sustainability entitled "Active Power Loss Reduction for Radial Distribution Systems by Placing Capacitors and PV systems", using the Stochastic Fractal Search Optimization Algorithm to determine the best size and the most appropriate sitting of capacitors and PV systems in radial distribution networks. They conclude that combination of capacitors and the PV system can reduce total power loss and improve voltage profile effectively. The introduction is well written and their findings are interesting. My only concern is this manuscript is a little bit too long. Other than that, this work can be accepted for publication.

Author Response

(The authors gave the same response as above.)

Round 2

Reviewer 1 Report

The authors addressed all my comments and presented the limitations in the revision. I have no further comments.